# Endogenous metabolites of vitamin E limit inflammation by targeting 5-lipoxygenase

Helmut Pein[1], Alexia Ville[2], Simona Pace[1], Veronika Temml[3], Ulrike Garscha[1], Martin Raasch[4], Khaled Alsabil[2], Guillaume Viault[2], Chau-Phi Dinh[2], David Guilet[2], Fabiana Troisi[1], Konstantin Neukirch[1], Stefanie König[1], Rosella Bilancia[5], Birgit Waltenberger[3], Hermann Stuppner[3], Maria Wallert[6], Stefan Lorkowski[6,7], Christina Weinigel[8], Silke Rummler[8], Marc Birringer[9], Fiorentina Roviezzo[5], Lidia Sautebin[5], Jean-Jacques Helesbeux[2], Denis Séraphin[2], Alexander S. Mosig [4], Daniela Schuster[10,11], Antonietta Rossi[5], Pascal Richomme[2], Oliver Werz[1] & Andreas Koeberle [1]

Systemic vitamin E metabolites have been proposed as signaling molecules, but their physiological role is unknown. Here we show, by library screening of potential human vitamin E metabolites, that long-chain ω-carboxylates are potent allosteric inhibitors of 5-lipoxygenase, a key enzyme in the biosynthesis of chemoattractant and vasoactive leukotrienes. 13-((2R)-6-hydroxy-2,5,7,8-tetramethylchroman-2-yl)-2,6,10-trimethyltridecanoic acid (α-T-13′-COOH) can be synthesized from α-tocopherol in a human liver-on-chip, and is detected in human and mouse plasma at concentrations (8–49 nM) that inhibit 5-lipoxygenase in human leukocytes. α-T-13′-COOH accumulates in immune cells and inflamed murine exudates, selectively inhibits the biosynthesis of 5-lipoxygenase-derived lipid mediators in vitro and in vivo, and efficiently suppresses inflammation and bronchial hyper-reactivity in mouse models of peritonitis and asthma. Together, our data suggest that the immune regulatory and anti-inflammatory functions of α-tocopherol depend on its endogenous metabolite α-T-13′-COOH, potentially through inhibiting 5-lipoxygenase in immune cells.

[1] Chair of Pharmaceutical/Medicinal Chemistry, Institute of Pharmacy, Friedrich-Schiller-University Jena, 07743 Jena, Germany. [2] Substances d'Origine Naturelle et Analogues Structuraux, SONAS, SFR4207 QUASAV, UNIV Angers, Université Bretagne Loire, 49070 Beaucouzé, France. [3] Institute of Pharmacy/Pharmacognosy and Center for Molecular Biosciences Innsbruck (CMBI), University of Innsbruck, 6020 Innsbruck, Austria. [4] Institute of Biochemistry II and Center for Sepsis Control and Care, University Hospital Jena, 07743 Jena, Germany. [5] Department of Pharmacy, School of Medicine, University of Naples Federico II, 80131 Naples, Italy. [6] Chair of Nutritional Biochemistry and Physiology, Institute of Nutrition, Friedrich-Schiller-University Jena, 07743 Jena, Germany. [7] Competence Cluster of Nutrition and Cardiovascular Health (nutriCARD), Halle, Jena and Leipzig, Jena 07743, Germany. [8] Institute of Transfusion Medicine, University Hospital Jena, 07747 Jena, Germany. [9] Department of Nutritional, Food and Consumer Sciences, Fulda University of Applied Sciences, 36037 Fulda, Germany. [10] Institute of Pharmacy/Pharmaceutical Chemistry and Center for Molecular Biosciences Innsbruck (CMBI), University of Innsbruck, 6020 Innsbruck, Austria. [11] Department of Pharmaceutical and Medicinal Chemistry, Institute of Pharmacy, Paracelsus Medical University Salzburg, 5020 Salzburg, Austria. Correspondence and requests for materials should be addressed to O.W. (email: oliver.werz@uni-jena.de) or to A.K. (email: andreas.koeberle@uni-jena.de)

Vitamin E, a mixture of eight natural vitamers (Supplementary Fig. 1a), comprises vital fat-soluble antioxidants that prevent damage to cellular lipids but also display anti-inflammatory activities independent of antioxidant properties[1,2]. Vitamin E deficiency is causative for severe degenerative diseases, decline of the immune response, and potentially for atherosclerosis[1,3]. Supplementation of α-tocopherol (α-T, 1a) and other vitamin E forms above the recommended daily dose showed marked anti-inflammatory, anti-atherosclerotic, and anti-tumor efficacy in animal models[1,2]. However, human clinical intervention studies provided mixed results regarding efficacy and safety of α-T (1a)[2,4,5]. The interest in vitamin E was recently renewed based on placebo-controlled, randomized, multicenter trials with positive outcomes for non-alcoholic fatty liver disease (NAFLD)[6,7] and Alzheimer disease[8]. Up to now, the molecular mechanism(s) and target(s) underlying the disease-modifying activities of vitamin E are still diffuse.

We here provide evidence that physiologically formed metabolites of α-T (1a) suppress inflammation by targeting 5-lipoxygenase (5-LO), which catalyzes the initial steps in the biosynthesis of potent immunomodulatory lipid mediators[9] (Supplementary Fig. 1b). 5-LO-derived leukotrienes (LTs) are pro-inflammatory key players in asthma and allergic rhinitis, trigger stress-induced oxidative DNA damage[10], and contribute to cardiovascular disease, inflammatory liver diseases, dermatitis, neurodegenerative disorders, and cancer[11–15]. Moreover, 5-LO participates in the biosynthesis of specialized pro-resolving mediators, such as lipoxins and resolvins that drive resolution physiology and tissue regeneration[16,17]. Upon release of arachidonic acid (AA) from phospholipids by cytosolic phospholipase $A_2$ (cPLA$_2$), AA is transferred by the membrane-bound 5-LO-activating protein (FLAP) to 5-LO that produces 5-hydro(pero)xy-eicosatetraenoic acid (5-H(P)ETE) and subsequently LTA$_4$ (Supplementary Fig. 1b). This intermediate is then converted to either LTB$_4$, a potent chemoattractant, or cysteinyl-LTs (LTC$_4$, LTD$_4$, LTE$_4$), which mediate smooth muscle contraction and enhance vascular permeability[11,18,19].

5-LO has been discussed as potential target of vitamin E before[20,21]. Thus, Goetzl reported in 1980 that the most abundant vitamin E form in mammalians, α-T (1a), modulates the lipoxygenation of AA in leukocytes[20]. Other forms of vitamin E were described to inhibit cellular LT formation in vitro by suppressing $Ca^{2+}$-influx, extracellular signal-regulated kinase (ERK)1/2 signaling and movement of 5-LO to FLAP[22]. However, the concentrations of the vitamin E forms required to suppress 5-LO product formation are not reached in human plasma[23], and it has been speculated that endogenous vitamin E is biotransformed to active metabolites, as established for other fat-soluble vitamins[24].

Hepatic ω- and β-oxidation of vitamin E produces long-chain metabolites (LCMs), which differ in oxygenation (ω-alcohol vs. ω-carboxylate), saturation, and length of the phytyl side chain as well as in methylation and phase II metabolism of the chromanol core[2] (Supplementary Fig. 1a). While the importance of bioactive metabolites is unquestioned for vitamins A and D, vitamin E-derived LCMs have only recently been proposed as signaling molecules[2]. δ-Carboxychromanols have been exemplarily investigated and described to inhibit 5-LO[22,25] and cyclooxygenase (COX)[26], another key enzyme of pro-inflammatory eicosanoid biosynthesis. Since these metabolites were detected neither in human nor rodent plasma unless vitamin E was supplemented at pharmacological doses[27,28], the physiological role of LCMs and their mode of action remained enigmatic. Two LCMs derived from α-T (1a) were recently discovered that occur at low nanomolar concentrations in human plasma[27,29] and inhibit pro-inflammatory signal transduction at supra-physiological LCM concentrations[2,30]. Here, we identify 13-((2R)-6-hydroxy-2,5,7,8-tetramethylchroman-2-yl)-2,6,10-trimethyltridecanoic acid (α-13′-carboxychromanol, α-T-13′-COOH, 4a) as allosteric and selective high-affinity inhibitor of 5-LO, which is substantially produced in a human liver organoid and reaches plasma concentration in humans and mice at which LT formation in human leukocytes is efficiently inhibited (Supplementary Fig. 1b). Moreover, α-T-13′-COOH (4a) accumulates in leukocytes and at sites of inflammation and suppresses acute inflammation in murine peritonitis. Conclusively, our data strongly suggest that α-T-13′-COOH (4a) is a highly potent bioactive metabolite of α-T (1a), which essentially contributes to the anti-inflammatory properties of α-T (1a) by targeting 5-LO.

## Results

**Vitamin E metabolites inhibit human 5-LO**. In order to thoroughly investigate metabolites of vitamin E forms, we first constructed a library based on an original set of ω-oxidized tocotrienols (TEs) isolated from *Garcinia amplexicaulis* (Clusiaceae). This comprehensive library was complemented by semi-synthesized analogs (see Supplementary Note 1) and contains intermediates of human vitamin E metabolism. Screening of this library for inhibition of 5-LO revealed high-affinity inhibitors among ω-alcohols and ω-carboxylates, including T and TE metabolites of the α-, β-, γ-, and δ-series, with the ω-carboxylated δ-TE (δ-TE-13′-COOH, 4h) being most potent (IC$_{50}$ = 35 nM) (Table 1). Metabolites with truncated side chain (5′-COOH and 3′-COOH, produced by repeated cycles of β-oxidation) markedly lost 5-LO inhibitory activity (IC$_{50}$ > 3 μM, except α-T-5′-COOH (5a), IC$_{50}$ = 750 nM for human recombinant 5-LO). The proposed intermediate LCM of mammalian hepatic δ-TE (1h) metabolism δ-tocodienol (DE)-13′-COOH (4l) lacks the Δ11′-double bond of TEs (Fig. 1a)[31] and is consequently less potent than the corresponding TE derivative (δ-TE-13′-COOH; 4h) in inhibiting 5-LO (IC$_{50}$ = 170 nM; Table 2). Cis–trans Δ11′ isomers (δ-TE-12a′-CH$_2$OH, 3h, vs. δ-TE-13′-CH$_2$OH, 2h) were equally active. Non-specific nuisance inhibition due to formation of detergent-sensitive colloid-like aggregates was excluded since the detergent Triton X-100 neither substantially affected 5-LO inhibition by δ-TE-13′-COOH (4h) nor by α-T (1a) (Fig. 1b). The 5-LO inhibitor zileuton was used as reference and suppressed 5-LO activity as expected (IC$_{50}$ = 840 nM).

**LCMs of vitamin E are allosteric 5-LO inhibitors**. 5-LO inhibitors encompass substrate-competitive, redox-active, and iron-chelating compounds[32] that occupy the AA-binding site and block substrate access or interfere with the active-site iron. An additional class of 5-LO inhibitors act at the C2-like domain, thereby preventing 5-LO translocation and binding to the nuclear membrane. As shown for δ-TE-13′-COOH (4h) as representative, LCMs inhibit 5-LO independent of the AA concentration (Fig. 1c) in a reversible manner (Fig. 1d).

Binding and direct interaction of LCMs with 5-LO was confirmed by a pull-down experiment, for which the carboxylic acid group of δ-TE-13′-COOH (4h) was amide-coupled to an amino-functionalized resin (Fig. 1e). Note that bulky amides such as δ-TE-13′-CONHBz (7) inhibit 5-LO comparably to acidic LCMs (Fig. 1a and Table 2). Excess of either AA or α-T (1a) did not prevent 5-LO binding to the δ-TE-13′-COOH (4h) construct (Fig. 1f), which either reflects striking differences in binding affinity or points towards separate binding sites for natural vitamin E forms and LCMs. Radical scavenging as measure for antioxidant activity was only evident at micromolar δ-TE-13′-COOH (4h) concentrations, far above the effective concentrations that inhibit 5-LO (Fig. 1g). Accordingly, a dimer of two LCMs was less active than the single moieties (Fig. 1a and Table 2).

**Table 1 Inhibitory activity of vitamin E derivatives against 5-LO in cell-free and cell-based test systems**

|  | T | | | | TE | | | |
|---|---|---|---|---|---|---|---|---|
|  | α (a) | β (b) | γ (c) | δ (d) | α (e) | β (f) | γ (g) | δ (h) |
| **Human recombinant 5-LO** | | | | | | | | |
| Vitamin E (**1**) | >1 | 0.75 ± 0.15 | 0.91 ± 0.15 | 0.31 ± 0.10 | 0.33 ± 0.08 | 0.19 ± 0.03 | 0.20 ± 0.06 | 0.17 ± 0.10 |
| 13′-CH₂OH (**2**) | 0.35 ± 0.04 | | | 0.12 ± 0.04 | 0.11 ± 0.01 | 0.09 ± 0.03 | | 0.15 ± 0.05 |
| 12′a-CH₂OH (**3**) | | | | | | | 0.12 ± 0.03 | 0.14 ± 0.03 |
| 13′-COOH (**4**) | 0.27 ± 0.01 | | | >1 | 0.46 ± 0.06 | 0.15 ± 0.04 | 0.30 ± 0.10 | 0.04 ± 0.02 |
| **Activated PMNL** | | | | | | | | |
| Vitamin E (**1**) | 808 ± 159 | 57 ± 2 | 502 ± 199 | 85 ± 5 | 277 ± 90 | 95 ± 5 | 132 ± 33 | 60 ± 15 |
| 13′-CH₂OH (**2**) | 0.19 ± 0.05 | | | 0.54 ± 0.02 | 0.27 ± 0.10 | 0.38 ± 0.09 | | 1.26 ± 0.33 |
| 12′a-CH₂OH (**3**) | | | | | | | 0.14 ± 0.02 | 0.22 ± 0.04 |
| 13′-COOH (**4**) | 0.08 ± 0.00 | | | 2.01 ± 0.59 | 1.70 ± 0.52 | 0.31 ± 0.11 | 0.49 ± 0.03 | 0.26 ± 0.09 |

$IC_{50}$ values (μM) are given as mean ± s.e.m.; $n = 3$ independent experiments.

To obtain insights into the 5-LO binding site of δ-TE-13′-COOH (**4h**), we combined molecular docking simulations with site-directed mutagenesis experiments. Docking scores are listed in Supplementary Table 1. High binding affinity of δ-TE-13′-COOH (**4h**) was predicted for a cavity between the regulatory C2-like and the catalytic domain of 5-LO, where hydrogen bonds to Arg101, Trp102 and Gln15 are potentially formed (Fig. 1h). Predicted binding patterns were similar for TEs, other LCMs and the amide δ-TE-13′-CONHBz (**7**), but were divergent and inconclusive for less potent Ts as well as side chain-truncated 3′- or 5′-carboxylates (Supplementary Fig. 2). The most active LCMs (IC₅₀ < 100 nM: δ-TE-13′-COOH, **4h**, β-TE-13′-CH₂OH, **2f**, δ-TE-13′-CONHBz, **7**) likely form hydrogen bonds at both ends of the cavity, namely between the farnesyl side chain and either Arg101, Gln141, or Val110 and the chromanol moiety and either Trp102, Asp170, or Gln15. While Trp102 is conserved in LO isoenzymes, Gln15 is unique for 5-LO, and Arg101 is present only in 5-LO, 12-LO-1, and 15-LO-1 but is replaced by glutamine in 12R-LO and 15-LO-2 (Supplementary Table 2).

The triple mutation of Trp102Ala/Trp13Ala/Trp75Ala or Arg101Asp replacement markedly altered the susceptibility of 5-LO towards δ-TE-13′-COOH (**4h**) (Fig. 1i). The Trp102Ala/Trp13Ala/Trp75Ala 5-LO mutant (5-LO_3W) was less potently inhibited versus wild-type enzyme, whereas inhibition of the Arg101Asp mutant was enhanced, despite disruption of the ionic interaction between the ω-carboxylate and Arg101. Docking studies suggest that the ionic interaction is not crucial for binding and that the carboxyl group is also stabilized by hydrogen bonding (Fig. 1h and Supplementary Fig. 2), with Lys133 and Gln141 residues in vicinity to the LCM carboxyl group functioning as alternative hydrogen bond partners. This finding is in line with our structure-activity relationship studies, which demonstrate that an anionic ω-carboxylate is not essential for 5-LO inhibition and can be replaced by non-ionic hydrogen acceptors such as hydroxyls or amides (Tables 1 and 2). Moreover, the positioning of δ-TE-13′-COOH (**4h**) in 5-LO rather precludes a redox-type mode of enzyme inhibition since the distance between the redox-active chromanol and the active site iron is too far for direct electron transfer, and also the existence of an electron transport chain is not obvious from the structure.

The proposed binding pocket for LCMs partially overlaps with the membrane binding site at the 5-LO C2-like domain that associates with phosphatidylcholine[18]. Whether δ-TE-13′-COOH (**4h**) competes with membrane phospholipids for 5-LO binding was investigated by phosphatidylcholine supplementation. Addition of phosphatidylcholine significantly impaired 5-LO

inhibition by δ-TE-13′-COOH (**4h**) in contrast to phosphatidylethanolamine (Fig. 1j), which does not specifically interact with 5-LO[18]. Together, our data suggest that δ-TE-13′-COOH (**4h**) targets a so far unexploited cavity that might represent a potential allosteric binding site for non-competitive 5-LO inhibitors.

**Occurrence of α-T-13′-COOH (4a) in humans and mice.** The diversity of LCMs in vivo is poorly investigated and limited to the analysis of plasma[27] and liver[28]. To quantify vitamin E metabolites in plasma and tissues, we developed a sensitive ultra-performance liquid chromatography ESI tandem mass spectrometry (UPLC-MS/MS)-based quantitation method. LCMs were extracted from biological samples by successive addition of methanol and chloroform (recovery rate: 63% for α-T-13′-COOH, **4a**), separated by C18-RP UPLC and detected by ESI-MS/MS through multiple reaction monitoring in the negative ionization mode. The 16 LCMs were quantified based on external calibration (eight standards) in combination with δ-TE-13′-COOH (**4h**) as internal standard. The detection response of α-T-13′-COOH (**4a**) was linear between 1 and 250 nM (Fig. 2a), with lower limits of detection between 0.5 and 4 nM for ω-carboxylates and 4 and 250 nM for ω-alcohols (Supplementary Table 3).

Analysis of plasma from ten healthy volunteers revealed α-T-13′-COOH (**4a**) as major systemic vitamin E metabolite (8–49 nM) (Fig. 2a, b) in accordance with previous studies[27,30]. Sex differences were not observed (Fig. 2b). Other ω-carboxylates than α-T-13′-COOH (**4a**) or ω-alcohols, for which the method is less sensitive (Supplementary Table 3), were below the detection limit. Mice, challenged with zymosan (i.p.) to develop acute peritonitis, exhibited comparable concentrations of circulating α-T-13′-COOH (**4a**) in plasma as human volunteers (Fig. 2c). Moreover, we detected significant amounts of carboxychromanols with shorter phytyl chain length in mice (α-T-11′-COOH, **11a**, α-T-9′-COOH, **12a**, and α-T-7′-COOH, **13a**). Most remarkable, ω-carboxylates are strongly enriched in the peritoneal exudates at sites of inflammation, with α-T-13′-COOH (**4a**) reaching concentrations up to 200 nM at 30 min post zymosan injection (Fig. 2d and Supplementary Table 4). Their levels rapidly decreased again within 4 h (Fig. 2e).

Oral administration of α-T (**1a**; 50 mg/kg) to mice significantly increased α-LCM levels in the exudates when given before immune cells infiltrate into the peritoneal cavity (Fig. 2d, e) with minor enrichment in plasma (Fig. 2c). Administration of α-T (**1a**) at a later time point was without effect on exudate LCMs (Supplementary Fig. 3). Co-administration of sesamin, an inhibitor of hepatic vitamin E catabolism that blocks CYP4F2/CYP3A4-dependent ω-hydroxylation[33], significantly impaired α-

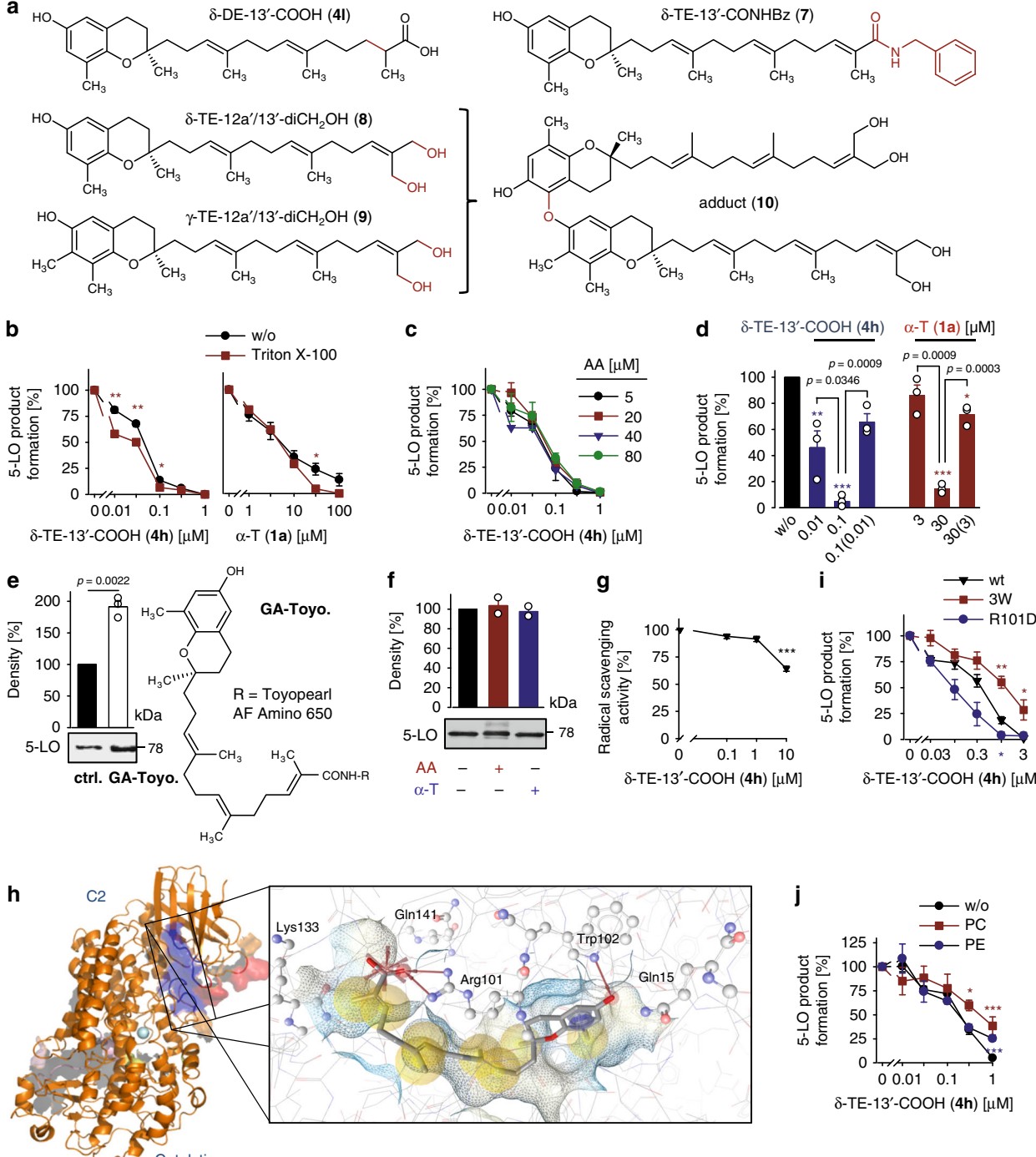

**Fig. 1** Molecular insights into 5-LO inhibition by LCMs. **a** Structures of LCMs. **b, c** Effect of Triton X-100 (0.01%, **c**) and of the substrate concentration (AA, **c**) on 5-LO inhibition by δ-TE-13′-COOH (**4h**) or α-T (**1a**). **d** Reversibility of 5-LO inhibition by δ-TE-13′-COOH (**4h**) and α-T (**1a**). Samples were pre-incubated with vehicle or compound for 15 min, 10-fold diluted and incubated for another 5 min before AA was added. Numbers in brackets indicate the diluted compound concentration after pre-incubation. **e, f** 5-LO pull-down by immobilized δ-TE-13′-COOH (**4h**; GA-Toyo) or immobilized 4-phenoxy butyric acid (ctrl.), which resembles a truncated form of δ-TE-13′-COOH (**4h**), in the presence of AA or α-T (**1a**; 100 μM, each, **f**). **g** Scavenging of DPPH radicals by δ-TE-13′-COOH (**4h**). **h** Molecular docking simulations of 5-LO showing the interaction of δ-TE-13′-COOH (**4h**) with Arg101 and Trp102 at the interface of the catalytic and regulatory C2-like domains. **i** Effect of δ-TE-13′-COOH (**4h**) on the inhibition of 5-LO product formation by HEK-293 cells that stably express either wild-type 5-LO (wt), triple 5-LO mutant Trp13Ala, Trp75Ala, and Trp102Ala (3 W) or Arg101Asp 5-LO. **j** Inhibition of purified 5-LO by δ-TE-13′-COOH (**4h**) in the presence of phosphatidylcholine (PC) or phosphatidylethanolamine (PE; 100 μg/ml, each). Mean ± s.e.m. (**b, c, g, i, j**) and single data (**d-f**) from $n = 3$ (**b-e, g, i, j**), $n = 2$ (**f**) independent experiments. *$P < 0.05$, **$P < 0.01$, ***$P < 0.001$ vs. absence of Triton X-100 (**b**), 20 μM AA (**c**), vehicle/control (**d-g**), wt 5-LO (**i**) or absence of phospholipids (**j**); two-tailed paired Student $t$ test (**b-e, i, j**) or repeated measures one-way ANOVA + Tukey HSD post hoc tests (**d, f, g**)

**Table 2 IC$_{50}$ values of LCMs for inhibition of purified human recombinant 5-LO**

| Compound | IC$_{50}$ (5-LO) ($\mu$M) |
|---|---|
| $\delta$-TE-13′-COOH (**4h**) | 0.04 ± 0.02 |
| $\delta$-DE-13′-COOH (**4l**) | 0.17 ± 0.04 |
| $\delta$-TE-13′-CONHBz (**7**) | 0.05 ± 0.02 |
| $\delta$-TE-12a′/13′-diCH$_2$OH (**8**) | 0.15 ± 0.04 |
| $\gamma$-TE-12′a/13′-diCH$_2$OH (**9**) | 0.11 ± 0.00 |
| Adduct (**10**) | 0.40 ± 0.11 |

IC$_{50}$ values ($\mu$M) are given as mean ± s.e.m.; $n = 3$ independent experiments.

T-13′-COOH (**4a**) levels in plasma but not in exudates (Fig. 2f, g), pointing towards α-T-13′-COOH (**4a**) pools with different metabolic fates. Together, α-T-13′-COOH (**4a**) is present in human and murine plasma, accumulates at sites of inflammation, and reaches concentrations in vivo at which 5-LO is effectively inhibited in vitro.

**Hepatic conversion of α-T (1a) to α-T-13′-COOH (4a).** Thus far, vitamin E metabolism was essentially studied in hepatocarcinoma cell line cultures that have limited predictive power for in vivo catabolism[2]. To identify metabolites that are produced from α-T (**1a**) in human liver, we used an in vitro organoid model of the human liver sinusoid. The liver-on-chip consists of two porous membranes serving as cell substrate and scaffold. The vascular layer resembles the endothelial lining of the liver sinusoid and comprises human umbilical vein endothelial cells (HUVEC) co-cultured with primary human monocyte-derived macrophages that mimic the functionality of immunomodulatory Kupffer cells (Supplementary Fig. 4a, b).

The hepatic compartment is formed by two hepatic layers aligned to each other, which contain HepaRG cells differentiated to hepatocyte-like and biliary-like cells that are co-cultured with LX-2 cells (stellate cell surrogates)[34]. HepaRG cells are considered as superior surrogate for primary human hepatocytes among hepatic cell lines. Kinetic studies on hepatocytes revealed peak levels of α-T-13′-COOH (**4a**) 48 h after addition of α-T (**1a**; 100 μM; physiological concentration of α-T, **1a**, in the liver) as determined by UPLC-MS/MS (Supplemental Fig. 4c). Under these conditions, the human liver organoid produces ω-carboxylates with varying chain length and α-T-13′-COOH (**4a**) as one of the major metabolites (Fig. 2h, i). Inhibition of CYP450 enzymes by sesamin suppressed the formation of α-T (**1a**) metabolites reaching significance for α-T-13′-COOH (**4a**) (Fig. 2j). Autoxidation of α-T (**1a**) to the 13′-carboxylate (**4a**) was below the detection limit in the absence of cells.

**Accumulating α-T-13′-COOH (4a) inhibits 5-LO in immune cells.** Natural forms of vitamin E suppress 5-LO activity in activated human polymorphonuclear leukocytes (PMNL) less potent (one to three magnitudes) than isolated 5-LO, whereas the inhibitory activity of LCMs was either slightly decreased (δ- and β-series and α-TEs), hardly affected (γ-TEs) or even markedly increased (α-Ts) in leukocytes (Table 1). Thus, α-T-13′-COOH (**4a**) suppressed 5-LO product formation in human PMNL (Fig. 3a, b) and monocytes (Fig. 3c) more potently than δ-TE-13′-COOH (**4h;** the most active LCM on isolated 5-LO), and inhibited LTB$_4$ formation in activated human blood at physiologically relevant in vivo concentrations (100 nM) (Fig. 3d). The higher potency in cellular versus cell-free assays prompted us to investigate whether α-T-13′-COOH (**4a**) targets additional factors in cellular 5-LO product biosynthesis. First, we assessed the viability

of peripheral blood mononuclear cells (PBMC) up to 24 h. Cell viability was not reduced by LCMs, except for the 12a′-CH$_2$OH series, albeit at high concentrations (10 μM) only (Table 3).

Impaired AA substrate release as a reason for superior effectiveness against 5-LO in intact cells can also be excluded because both α-T-13′-COOH (**4a**) and δ-TE-13′-COOH (**4h**) consistently inhibited 5-LO also when exogenous AA was supplied (Fig. 3a, b). Moreover, α-T-13′-COOH (**4a**) neither affected intracellular Ca$^{2+}$, an essential cofactor for cellular 5-LO activation (Fig. 3e), nor interrupted the 5-LO/FLAP complex where FLAP transfers AA to 5-LO[11,18,19] (Fig. 3f). While similar results were obtained for δ-TE-13′-COOH (**4h**) at submicromolar concentrations, intracellular Ca$^{2+}$-influx was significantly reduced at ≥1 μM (Fig. 3e, f).

Finally, we investigated whether α-T-13′-COOH (**4a**) selectively enriches in leukocytes. In fact, human PMNL preferentially accumulate α-T-13′-COOH (**4a**) over δ-TE-13′-COOH (**4h**), reaching micromolar intracellular concentrations when assuming a spherical shape of PMNL with an average diameter of 13 μm (Fig. 3g). The dramatic enrichment of α-T-13′-COOH (**4a**) in leukocytes thus provides an explanation for the superior inhibition of 5-LO compared to cell-free assays. Accordingly, the concentrations of α-T-13′-COOH (**4a**), α-T-11′-COOH (**11a**) and α-T-9′-COOH (**12a**) are substantially higher in peritoneal leukocytes (Fig. 3h) compared to the surrounding exudate in zymosan-challenged mice (Fig. 2e and Supplementary Fig. 3).

**Multiple subordinated targets of α-T-13′-COOH (4a).** The biosynthesis of eicosanoids and docosanoids depends on PLAs, LOs, COX, and CYP450 enzymes, which produce a variety of lipid mediators that critically regulate inflammation as well as resolution[9,16]. At higher concentrations, α-T-13′-COOH (**4a**) significantly impaired 12-LO (IC$_{50}$ > 3 μM) (Fig. 4a) and nonsignificantly 15-LO product formation in activated human PMNL (Fig. 4b). Other LCMs inhibited 12- and 15-LO less efficient and some even enhanced the generation of 15-LO products (Table 3). Moreover, α-T-13′-COOH (**4a**) inhibited (i) isolated bovine COX-1 (IC$_{50}$ > 10 μM) (Fig. 4c) and COX-1 in human platelets (IC$_{50}$ = 6.9 μM) (Fig. 4d), (ii) human recombinant LTC$_4$ synthase (LTC$_4$S) (IC$_{50}$ = 9.9 μM) (Fig. 4e), (iii) human microsomal prostaglandin E$_2$ synthase (mPGES)-1 in microsomal preparations (IC$_{50}$ = 8.8 μM) (Fig. 4f), and (iv) human recombinant soluble epoxide hydrolase (sEH; IC$_{50}$ > 10 μM) (Fig. 4g). In contrast, human recombinant COX-2 (Fig. 4h) and COX-2-dependent prostaglandin (PG)E$_2$ formation in activated human monocytes (Fig. 4i) were not inhibited up to 10 μM. Conclusively, α-T-13′-COOH (**4a**) potently inhibits 5-LO (high affinity target) but hardly other enzymes within eicosanoid biosynthesis (low affinity targets). The consequences on the cellular lipid mediator network were investigated in activated human monocytes by comprehensive profiling of eicosanoids and docosanoids using UPLC-MS/MS. α-T-13′-COOH (**4a**) inhibited the biosynthesis of 5-LO-derived products, i.e., LTB$_4$, its isomers, and 5-hydroxy-eicosatetraenoic acid (5-HETE), but neither markedly suppressed the formation of COX-derived PGs, 12- and 15-LO products nor other lipid mediators (Fig. 4j). Thus, the effect of α-T-13′-COOH (**4a**) on the cellular lipid mediator profile is determined by 5-LO inhibition. A similar profile was observed for δ-TE-13′-COOH (**4h**) (Fig. 4a–i and Supplementary Fig. 5).

**α-T-13′-COOH (4a) limits inflammation in murine peritonitis.** The physiological relevance of 5-LO inhibition by α-T-13′-COOH (**4a**) was investigated for zymosan-induced murine peritonitis in vivo. In this model, LTs and other lipid mediators critically contribute to initiation and progression of

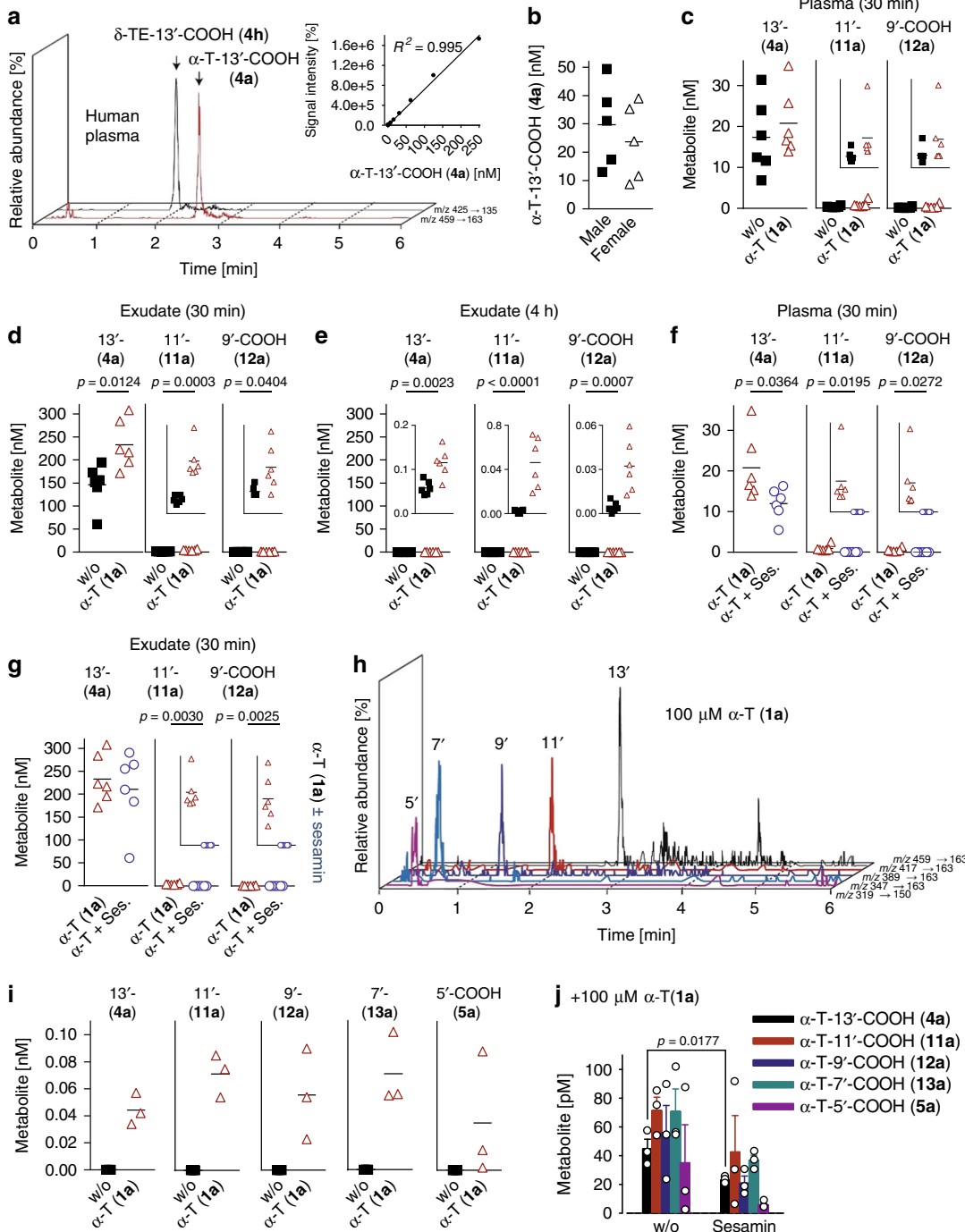

**Fig. 2** α-T-13′-COOH (**4a**) is present in plasma, enriched at inflammatory sites and produced from α-T by a liver-on-chip. **a** Extracted chromatograms of α-T-13′-COOH (**4a**) and the internal standard δ-TE-13′-COOH (**4h**) detected in plasma from healthy donors. Insert: external calibration curve for α-T-13′-COOH (**4a**). The $R^2$ value indicates the correlation coefficient obtained after carrying out linear regression analysis. **b** Concentrations of α-T-13′-COOH (**4a**) in male and female human plasma. **c–g** Concentration of α-T-13′-COOH (**4a**), α-T-11′-COOH (**11a**) and α-T-9′-COOH (**12a**) in plasma (**c**, **f**) and peritoneal exudates (**d**, **e**, **g**) of mice with acute peritonitis 30 min (**c**, **d**, **f**, **g**) or 4 h (**e**) post zymosan injection. Vehicle (DMSO; "w/o"), α-T (**1a**; 50 mg/kg, p.o.) and sesamin (250 mg/kg, p.o.; "Ses.") were given 60 min prior to zymosan injection. The inserts zoom in on the data. **h**, **i** Extracted chromatograms (**h**) and concentrations (**i**) of LCMs that were produced in HepaRG biochips (**h**) and biochip-based human liver organoids (liver-on-chips; **i**) from α-T (**1a**; 100 μM) within 48 h. **j** Effect of sesamin (3 μM) on the conversion of α-T (**1a**; 100 μM) to LCMs by the liver-on-chip within 48 h. Mean with single data (**b–g**, **i**) or mean + s.e.m. (**j**) from $n = 5$ (**b**) donors, $n = 6$ (**c–g**) mice, $n = 3$ (**i**, **j**) independent experiments. **c**, **d**, **f**, **g**, **i**, and **j** in parts refer to the same control datasets. Two-tailed unpaired Student $t$ test

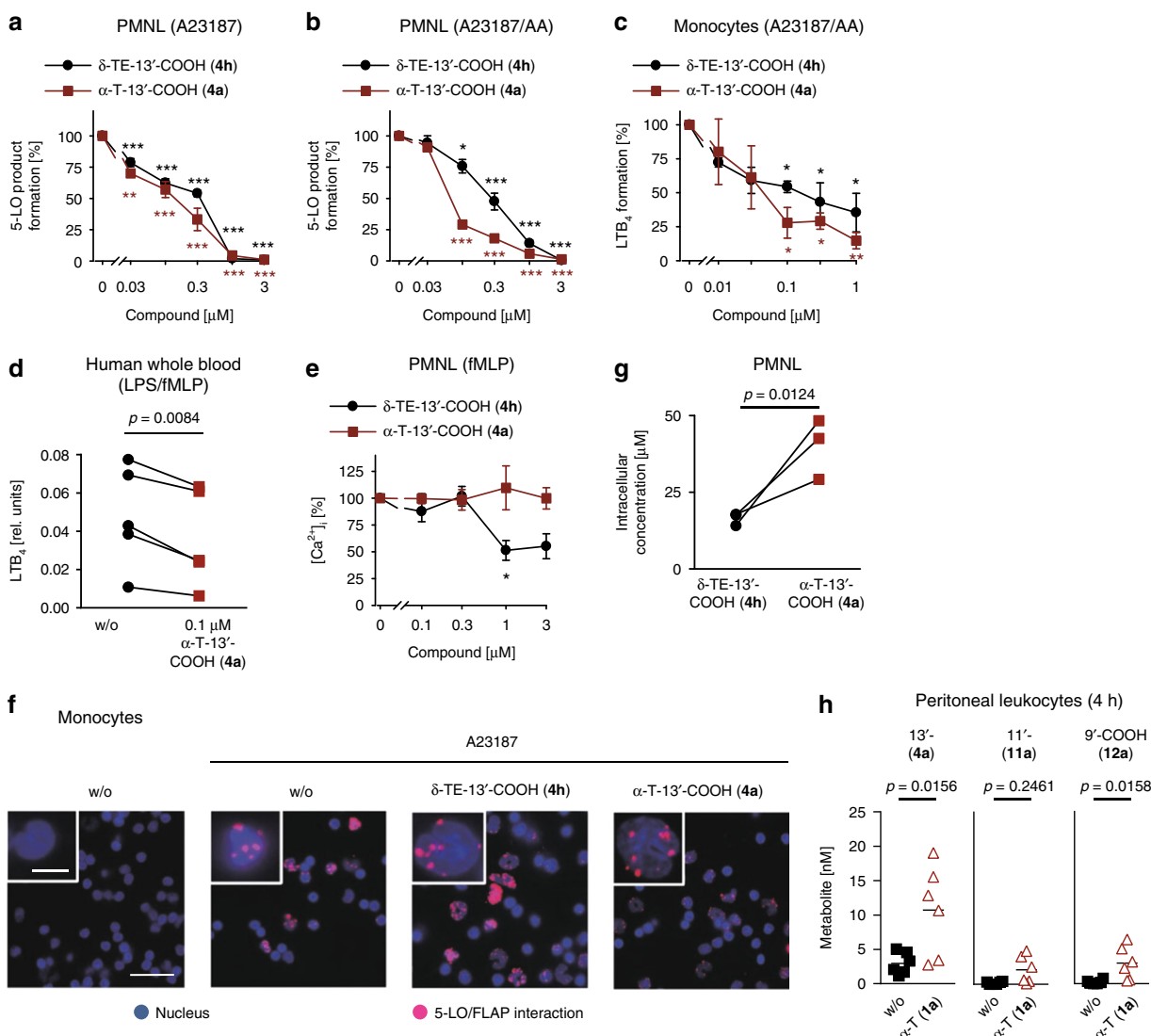

**Fig. 3** Accumulation of α-T-13′-COOH (**4a**) in immune cells favors inhibition of 5-LO. **a–c** Effects of α-T-13′-COOH (**4a**) and δ-TE-13′-COOH (**4h**) on the formation of 5-LO products (LTB$_4$, its isomers and 5-H(P)ETE analyzed by RP-HPLC) in PMNL treated with Ca$^{2+}$-ionophore A23187 (**a**) plus AA (**b**) and on LTB$_4$ formation (analyzed by UPLC-MS/MS) in A23187/AA-activated monocytes (**c**) and LPS/fMLP-stimulated human blood (**d**). **e** Intracellular Ca$^{2+}$ concentration ([Ca$^{2+}$]$_i$) in fMLP-stimulated PMNL treated with α-T-13′-COOH (**4a**) or δ-TE-13′-COOH (**4h**). **f** Influence of α-T-13′-COOH (**4a**) and δ-TE-13′-COOH (**4h**; 3 μM, each) on the formation of the 5-LO/FLAP complex in A23187-activated monocytes as determined by proximity ligation assay; scale bar of the outer box: 25 μm, scale bar of the insert: 5 μm. Images are representative of three independent experiments. **g** Uptake of α-T-13′-COOH (**4a**) and δ-TE-13′-COOH (**4h**; 150 nM, each) by PMNL, which were treated with either of the compounds for 20 min. Average intracellular concentrations were calculated for spherical PMNL with a diameter of 13 μm. Interconnected lines indicate data from the same independent experiment. **h** Accumulation of α-T-13′-COOH (**4a**) in leukocytes from peritoneal exudates of mice 4 h post zymosan injection. α-T (50 mg/kg) was administered p.o. 1 h before zymosan. Mean ± s.e.m. (**a–c**, **e**), paired data (**d**, **g**) or mean with single data (**h**) from $n = 3$ (**a–c**, **e**, **g**) or $n = 5$ (**d**) independent experiments, $n = 6$ (**h**) mice. *$P < 0.05$, **$P < 0.01$, ***$P < 0.001$ vs. vehicle control (**a–e**, **h**) or δ-TE-13′-COOH (**4h**)-treated cells (**g**); repeated measures one-way ANOVA + Tukey HSD post hoc tests (**a–c**, **e**), two-tailed paired Student $t$ test (**d**, **g**) or two-tailed unpaired Student $t$ test (**h**)

inflammation[35] (Fig. 5a). LTC$_4$ produced by resident peritoneal macrophages increases vasopermeability of postcapillary venules. Infiltrated neutrophils, generating the potent chemoattractant LTB$_4$, dominate the inflammatory reaction at later stages. α-T (**1a**) and α-T-13′-COOH (**4a**) were given i.p. to circumvent the hepatic first-pass metabolism to LCMs. Accordingly, the concentration of LCMs in plasma and exudate (3:1 distribution) was substantially increased only when α-T-13′-COOH (**4a**) was administered, whose degradation to short-chain metabolites was minute (Fig. 5b). α-T-13′-COOH (**4a**) reached substantially higher concentrations in plasma and exudate upon i.p. injection (1.4 and 3.9 μM, see Fig. 5b) than following oral administration of

α-T (**1a**; 21 and 233 nM, see Fig. 2c, d). LTC$_4$ levels (Fig. 5c) were strongly decreased along with reduced vascular permeability (Fig. 5d) as well as impaired LTB$_4$ levels (Fig. 5e) and cell infiltration (Fig. 5f), which is dominated by the influx of neutrophils[35]. α-T (**1a**) was significantly less effective in reducing exudate levels of LTC$_4$ (Fig. 5c). Moreover, lipidomic profiling of plasma and exudate from α-T-13′-COOH (**4a**)-treated mice revealed a preferential drop of 5-LO products among various lipid mediators as shown by UPLC-MS/MS (Fig. 5g) and confirmed by ELISA (Fig. 5h). Thus, α-T-13′-COOH (**4a**), which accumulates at sites of inflammation, possesses potent anti-inflammatory activity seemingly by suppressing LT formation. Interestingly, α-

**Table 3 Effects of vitamin E derivatives on 12- and 15-LO product formation in PMNL and on cell viability of PBMC**

| | Ts | | | | TEs | | | |
|---|---|---|---|---|---|---|---|---|
| | α (a) | β (b) | γ (c) | δ (d) | α (e) | β (f) | γ (g) | δ (h) |
| 12-LO product formation/PMNL^a | | | | | | | | |
| Vitamin E (**1**) | n.i. | n.i. | n.i. | n.i. | 88.9 ± 4.9 | 65.4 ± 6.1* | 70.8 ± 2.3* | 87.6 ± 8.5 |
| 13′-CH$_2$OH (**2**) | 84.7 ± 4.7 | | | 84.8 ± 7.8 | 62.7 ± 15.2 | 57.8 ± 13.6 | | n.i. |
| 12′a-CH$_2$OH (**3**) | | | | | | 83.5 ± 3.6 | 70.6 ± 6.3 | |
| 13′-COOH (**4**) | 55.3 ± 1.7*** | | | 74.5 ± 5.9* | 72.6 ± 13.6 | 70.0 ± 14.1 | n.i. | 75.1 ± 12.1 |
| 5′-COOH (**5**) | 56.7 ± 3.0 | | | | | | | |
| 3′-COOH (**6**) | 87.2 ± 3.6 | | 89.9 ± 4.2 | n.i. | | | | |
| 15-LO product formation/PMNL^a | | | | | | | | |
| Vitamin E (**1**) | n.i. | n.i. | n.i. | 86.6 ± 15.8 | n.i. | n.i. | n.i. | n.i. |
| 13′-CH$_2$OH (**2**) | 156.7 ± 26.2 | | | 142.8 ± 13.9 | 88.0 ± 17.0 | 78.9 ± 14.8 | | n.i. |
| 12′a-CH$_2$OH (**3**) | | | | | | | 168.8 ± 1.2*** | 174.6 ± 6.9* |
| 13′-COOH (**4**) | 52.6 ± 20.3 | | | 121.6 ± 10.9 | 181.5 ± 26.5 | 186.1 ± 35.8 | 169.2 ± 4.5** | 128.6 ± 42.3 |
| 5′-COOH (**5**) | n.i. | | | | | | | |
| 3′-COOH (**6**) | n.i. | | n.i. | n.i. | | | | |
| Cell viability/peripheral blood mononuclear cells^b | | | | | | | | |
| Vitamin E (**1**) | n.i. | n.i. | n.i. | n.i. | n.d. | n.d. | n.d. | n.i. |
| 13′-CH$_2$OH (**2**) | n.i. | | | n.i. | n.d. | n.d. | | n.i. |
| 12′a-CH$_2$OH (**3**) | | | | | | | n.i./43.7 ± 16.2* | n.i./73.5 ± 14.4 |
| 13′-COOH (**4**) | n.i. | | | n.i. | n.i. | n.i. | n.i. | n.i. |
| 5′-COOH (**5**) | n.d. | | | | | | | |
| 3′-COOH (**6**) | n.d. | | n.d. | n.d. | | | | |

Residual viability/activities (% of control) are given as mean ± s.e.m.; $n = 3$ independent experiments
n.i. no inhibition, n.d. not determined
*$P < 0.05$, **$P < 0.01$, ***$P < 0.001$ vs. vehicle control; two-tailed paired Student $t$ test. ^aAt 3 μM; ^bAt 1 and 10 μM, respectively

T-13′-COOH (**4a**) was more effective in reducing LT production and limiting inflammation than the clinically used 5-LO inhibitor zileuton at 30 min post zymosan injection (Fig. 5c, d), while this difference disappeared or was even reversed at the later time point (Fig. 5e, f). The superior effect of zileuton on immune cell recruitment (Fig. 5f) might be explained from the interference with a broad spectrum of lipid mediators besides 5-LO products and a redirection of the lipid mediator network (Fig. 5g).

The resolution phase of murine peritonitis was studied at 18 h after zymosan injection, which is the latest time point before peritoneal cell numbers readjust to baseline (Supplementary Fig. 6). The suppressive effect of α-T-13′-COOH (**4a**) on cell infiltration vanished at the resolution phase (18 h) (Fig. 5f). Lipid mediator profiles in the peritoneal exudates were partially counter-regulated (Supplementary Fig. 7), with both low abundant pro-inflammatory (COX- and 5-LO products) and pro-resolving mediators (protectins, maresins) slightly increasing (Fig. 5i). Similar changes in the lipid mediator profile were observed for zileuton. On the other hand, α-T-13′-COOH (**4a**, but not zileuton) strongly increased the circulating resolvin E3 in plasma (Fig. 5j) while the levels of other lipid mediators were either decreased or remained unchanged (Supplementary Fig. 7), which might hint towards a role of α-T-13′-COOH (**4a**) in systemic resolution.

**α-T-13′-COOH (4a) reduces airway hyper-reactivity.** Vitamin E status has been associated with prevalence of asthma in humans[2], where LTs play a fundamental role in pathogenesis, e.g., by inducing brochoconstriction[11]. Thus, we investigated the effect of α-T-13′-COOH (**4a**) in an experimental model of asthma. Mice were sensitized to ovalbumin (OVA), and bronchi were harvested to assess airway hyper-reactivity (Fig. 6a). Administration (i.p.) of α-T-13′-COOH (**4a**) lowered pulmonary LTC$_4$ to basal levels (Fig. 6b) and significantly inhibited OVA-induced bronchial hyper-reactivity to the cholinergic agonist carbachol (Fig. 6c),

which points toward a role of 5-LO inhibition by α-T-13′-COOH (**4a**) in limiting asthmatic airway contraction.

## Discussion

Here, we report on the discovery of a bioactive metabolite of vitamin E that is endogenously formed in humans and mice and may contribute to the anti-inflammatory properties of vitamin E as an allosteric high-affinity inhibitor of 5-LO (Supplementary Fig. 1). Our hypothesis is based on the following findings: (i) α-T-13′-COOH (**4a**) is endogenously formed in humans and mice, (ii) its biosynthesis is ascribed to hepatocytes in human liver organoids, (iii) its basal plasma concentration is relevant to regulate 5-LO activity, (iv) it strongly accumulates at sites of inflammation, (v) it selectively reduces 5-LO products in vitro and in vivo along with suppression of acute inflammation in murine peritonitis, and (vi) it reduces LT-mediated bronchial hyper-reactivity in an experimental model of asthma—one of the most relevant and validated LT-driven diseases[11,18]. It is tempting to speculate that α-T-13′-COOH (**4a**) regulates 5-LO activity in circulating immune cells depending on diet, metabolic turnover and inflammatory status.

The accumulation of α-T-13′-COOH (**4a**) in inflamed exudates from murine acute peritonitis correlates with substantial uptake of the LCM by PMNL, which infiltrate into the peritoneal cavity and represent major sources for 5-LO-derived LTs[11,35]. PMNL preferentially take up α-LCM, as reflected by superior cell-based 5-LO inhibitory activities versus isolated 5-LO, and confirmed for α-T-13′-COOH (**4a**) by analyzing intracellular LCM concentrations. On the one hand, the higher lipophilicity of α-T-13′-COOH (**4a**; calculated log $P = 8.81$) compared to δ-TE-13′-COOH (**4h**; calculated log $P = 6.65$) might facilitate passive membrane transfer. On the other hand, selectivity for α-T-13′-COOH (**4a**) might depend on specific uptake, cellular sorting or retention mechanisms, which are unknown for LCMs so far. The hepatic cytosolic α-tocopherol transfer protein (α-TPP)

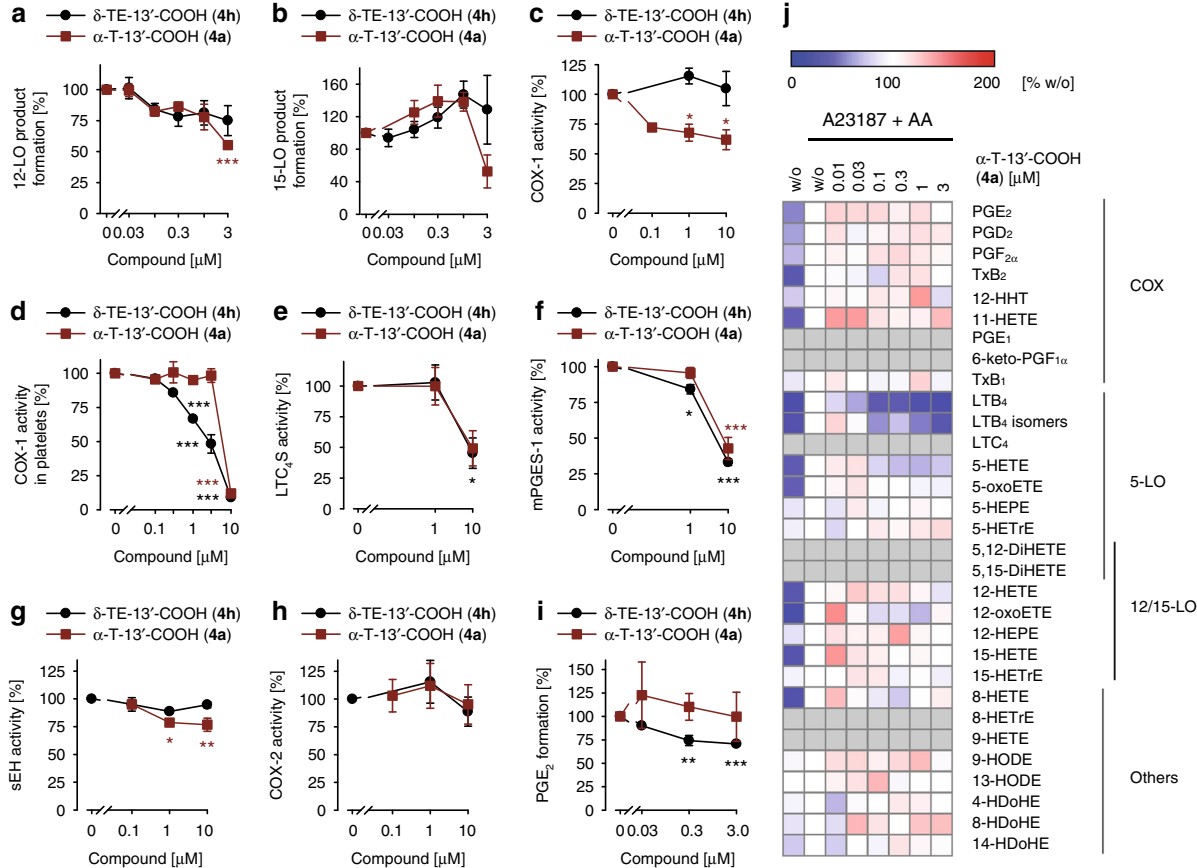

**Fig. 4** Moderate and subordinated effects of α-T-13′-COOH (**4a**) within eicosanoid and docosanoid biosynthesis. **a–i** Effects of α-T-13′-COOH (**4a**) and δ-TE-13′-COOH (**4h**) on the biosynthesis of 12- (**a**) and 15-LO-derived products in A23187/AA-activated PMNL (**b**), isolated bovine COX-1 (**c**), COX-1-dependent 12-HHT formation in platelets (**d**), human recombinant LTC₄ synthase (LTC₄S) (**e**), mPGES-1 in microsomal preparations of IL-1β-treated A549 cells (**f**), human recombinant sEH (**g**), human recombinant COX-2 (**h**), and COX-2-dependent PGE₂ formation in LPS-activated monocytes that were treated with A23187 plus AA (**i**). **j** Heatmap showing the concentration-dependent effect of α-T-13′-COOH (**4a**) on the lipid mediator profile in A23187/AA-treated monocytes that were pre-activated with LPS. Tx thromboxane, HTT hydroxy-heptadecatrienoic acid, oxoETE oxo-eicosatetraenoic acid, HEPE hydroxy-eicosapentaenoic acid, HETrE hydroxy-eicosatrienoic acid, (Di)HETE (di)hydroxy-eicosatetraenoic acid, HODE hydroxy-octadecadienoic acid, HDoHE hydroxy-docosahexaenoic acid. Mean ± s.e.m. (**a–i**) and mean (**j**) from n = 3 (**a–d**, **f–j**), n = 5 (**e**) independent experiments. *P < 0.05, **P < 0.01, ***P < 0.001 vs. vehicle control; repeated measures one-way ANOVA + Tukey HSD post hoc tests

binds to α-T (**1a**) with high affinity and facilitates the transfer into secreted plasma lipoproteins[3,36]. A similar mechanism might retain α-LCMs in PMNL and potentially other cell types.

Multiple LCMs (including δ-TE-13′-COOH, **4h**) inhibit 5-LO but only α-T-13′-COOH (**4a**[27]; and this study), α-T-13′-CH₂OH (**2a**)[29] and chain-shortened ω-carboxylates of α-T (**1a**; this study) have been detected in humans and rodents. The physiological relevance of other LCMs remains speculative in light of comparable detection limits for long-chain ω-carboxylates. γ- and δ-LCMs might gain importance when natural vitamin E forms are supplemented. In fact, γ-carboxychromanols are present in plasma of mice (up to 150 nM) receiving γ-T (**1c**)/γ-TE (**1g**)[31]. Conjugated LCMs, such as O-sulfates, even reach higher plasma levels but our docking studies rather preclude efficient 5-LO inhibition, at least by targeting the proposed LCM binding site. Along these lines, O-sulfates also failed to inhibit COX-2[26].

We reveal LCMs as reversible, non-competitive, allosteric inhibitors of human 5-LO and disprove previous speculations based on plant LOs that proposed an irreversible mode of inhibition for α-T (**1a**)[37]. Alternative mechanisms that regulate 5-LO product formation, i.e., the supply of AA, Ca²⁺-dependent nuclear translocation of 5-LO and interaction with FLAP, are seemingly not targeted by α-T-13′-COOH (**4a**). Moreover, LCMs

do not compete with AA at the active site but may preferentially bind to an allosteric pocket that partially overlaps with the membrane binding region of 5-LO, as exemplarily shown for δ-TE-13′-COOH (**4h**; δ-trans-tocotrienolic acid, garcinoic acid). Of interest, δ-TE-13′-COOH (**4h**) is a major secondary metabolite contained in the bitter nut *G. kola*—an anti-inflammatory, bronchodilatory and anti-infectious remedy in African traditional medicine[38]. The physiologically relevant LCM of δ-TE (**1h**), which has been proposed to be a tocodienol lacking the Δ11 double bond of TEs[31], less potently inhibits 5-LO when compared with δ-TE-13′-COOH (**4h**) (Table 2).

Interference with low-affinity targets of α-T-13′-COOH (**4a**), i.e., COX-1, mPGES-1, and 12-LO, does not markedly influence eicosanoid and docosanoid profiles in vitro and in vivo. However, it is likely that the weak interaction of α-T-13′-COOH (**4a**) with these targets buffers the redirection of AA when 5-LO is blocked and thus contributes to eicosanoid network stability as recently discussed[39,40]. Most remarkably, α-TE (**1e**) has been reported to inhibit porcine 12-LO thereby blocking glutamate-induced cell death at nanomolar concentrations in neuronal cells[41]. We confirmed the inhibition of human 12-LO in PMNL by TEs and identified LCMs with superior inhibitory activity, though the physiological relevance of 12-LO as target is contrasted by the

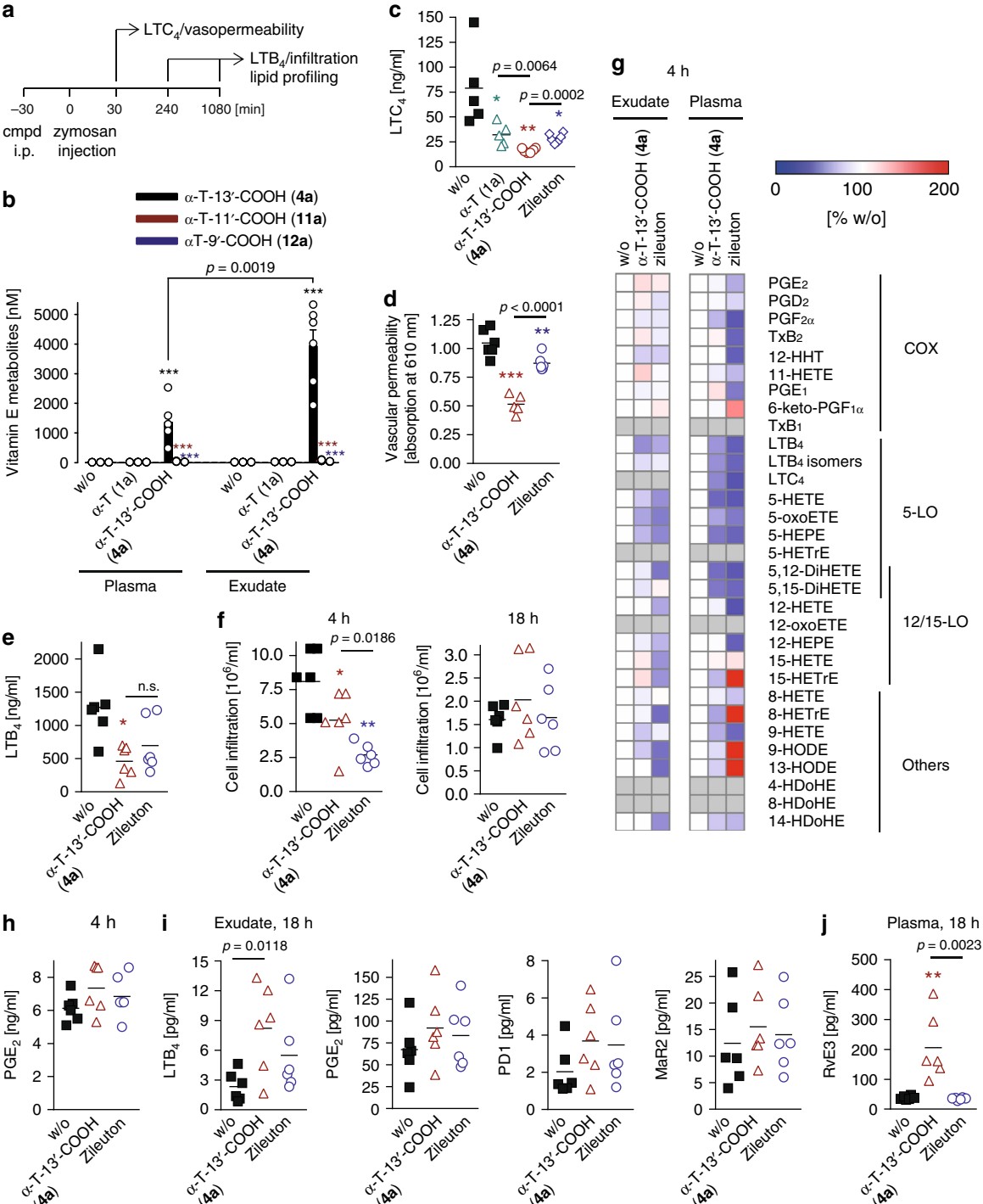

**Fig. 5** α-T-13′-COOH (**4a**) inhibits 5-LO product formation in vivo along with reduced inflammation in murine peritonitis. **a** Time-scale for zymosan-induced murine peritonitis. Cmpd, compounds. **b–j** Vehicle (DMSO), α-T (**1a**), α-T-13′-COOH (**4a**) or zileuton (10 mg/kg, i.p., each) were administered to mice, which were killed at 30 min (**c**, **d**), 240 min (**b**, **e–h**) or 18 h post zymosan injection (**f**, **i**, **j**) as indicated in **a**. **b** Concentration of α-LCMs in plasma and peritoneal exudate. **c** Levels of LTC$_4$ in the exudate determined by ELISA. **d** Vascular permeability. **e** Levels of LTB$_4$ in the exudate determined by ELISA. **f** Immune cell infiltration into the peritoneal cavity. **g** Lipid mediator profiles in the exudate and plasma analyzed by UPLC-MS/MS. Tx thromboxane, HTT hydroxy-heptadecatrienoic acid, oxoETE oxo-eicosatetraenoic acid, EPE hydroxy-eicosapentaenoic acid, HETrE hydroxy-eicosatrienoic acid, (Di)HETE (di) hydroxy-eicosatetraenoic acid, HODE hydroxy-octadecadienoic acid, HDoHE, hydroxy-docosahexaenoic acid. **h** Levels of PGE$_2$ in the exudate determined by ELISA. **i** Levels of LTB$_4$, PGE$_2$, protectin (P)D1 and maresin (MaR)2 in exudates analyzed by UPLC-MS/MS. **j** Levels of resolvin (Rv)E3 in plasma analyzed by UPLC-MS/MS. Mean + s.e.m. (**b**), mean with single data (**c–f**, **h–j**) or mean (**g**) from $n = 5$ (**c** for w/o, α-T, **1a**, α-T-13′-COOH, **4a**; **d** for α-T-13′-COOH, **4a**), $n = 6$ (**b**, **c** for zileuton, **d** for w/o, zileuton, **e–j**) mice. *$P < 0.05$, **$P < 0.01$, ***$P < 0.001$ vs. vehicle control; one-way ANOVA + Tukey HSD post hoc tests or two-tailed unpaired Student $t$-test

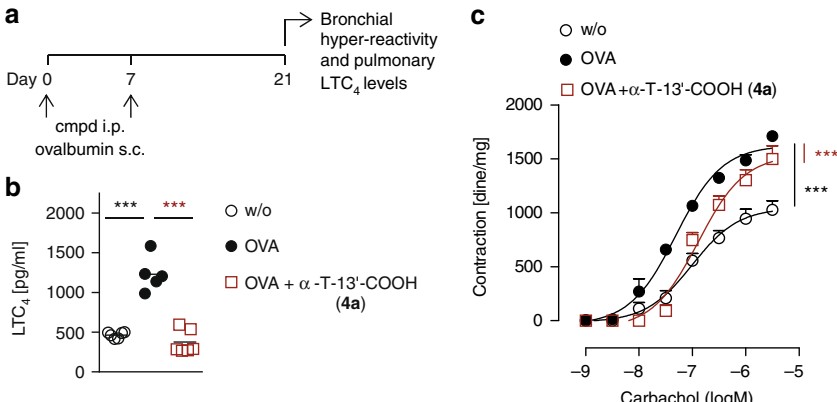

**Fig. 6** α-T-13′-COOH (**4a**) inhibits bronchial hyper-reactivity in OVA-sensitized mice. **a** Time-scale for sensitizing mice with OVA. Cmpd, compounds. **b**, **c** Vehicle (DMSO) or α-T-13′-COOH (**4a**) (10 mg/kg, i.p.) were given to mice prior to injection of OVA (100 μg adsorbed to 3.3 mg of aluminum hydroxide gel, s.c.) at days 0 and 7. Pulmonary LTC$_4$ levels (**b**) and bronchial reactivity to carbachol (**c**) were evaluated 21 days after administration of OVA. Mean + s.e.m. from $n = 10$ (w/o) or $n = 6$ (OVA, OVA + α-T-13′-COOH, **4a**) mice. ***$P < 0.001$ vs. OVA-treated control mice; two-way ANOVA + Bonferroni post hoc tests or two-tailed unpaired Student $t$-test

discrepancy between high effective concentrations (IC$_{50}$ > 3 μM) and low plasma levels of TEs (≤0.35 μM)[23] and LCMs (refs. [27,29] and this study) unless α-TE (**1e**) is supplemented[42].

Multiple, partly opposing molecular mechanisms have been reported for vitamin E forms and LCMs. Besides their well-established anti-oxidative properties, Ts, TEs, and LCMs differentially regulate gene expression of inflammatory, lipid metabolic, and neuroactive proteins, and they act on enzymes with important roles in immunoregulation[2]. For example, α-T (**1a**) inhibits protein kinase (PK)Cα by binding to its regulatory domain and activates phosphatase type 2A, which de-phosphorylates and thus inactivates protein PKCα[2], whereas γ-T (**1c**) induces PKCα activity and negatively correlates with lung function in humans[2]. Conclusively, the composition and local tissue concentration of vitamin E forms and their conversion to bioactive metabolites are important factors that might influence the clinical outcome of vitamin E supplementation studies.

Individual differences in vitamin E form uptake, distribution and metabolism are likely smaller in animal studies with defined diet and genetic background than in humans participating in clinical trials. Accordingly, animal studies showed clear benefits for α-T (**1a**) intake in diseases with inflammatory component, such as atherosclerosis, lung inflammation and cancer, whereas clinical vitamin E intervention studies often produced disappointing results or even questioned the safety of pharmacological doses of vitamin E (≥400 IU/d)[1,2,4,5]. Along these lines, vitamin E deficiency increased LT biosynthesis/turnover in humans and rats[43,44], whereas an excess of vitamin E produced contradictory results[2,4,5]. We hypothesize that individual differences in the capacities to biosynthesize LCMs essentially propagate such discrepancies, which become relevant at high doses of vitamin E and might have a strong impact on the clinical outcome.

5-LO generates bioactive lipid mediators that are involved in inflammatory and allergic reactions as well as in autoimmune diseases, and in fact, vitamin E intake has long been associated with the regulation of related diseases, including asthma, cardiovascular diseases, and Alzheimer's disease[1,2,45]. Moreover, LT formation in PMNL supports lung colonization of metastasis-initiating breast cancer cells in mouse models[12], providing a mechanistic basis for the well-established pro-tumor properties of 5-LO[18]. Remarkable are also the consistent outcomes of clinical studies addressing the therapeutic potential of vitamin E in NAFLD with manifestation of the metabolic syndrome[6,7],

because LCMs are strikingly enriched in liver upon oral administration of vitamin E as shown for γ-T (**1c**) in rats[28]. Moreover, metabolic dysregulations that promote NAFLD led to reduced hepatic expression of CYP4F2, the initial enzyme in LCM biosynthesis, in a high fat murine model of steatosis[46], and to lower excretion of vitamin E metabolites by patients with metabolic syndrome[47]. In consequence, the progression from healthy liver to non-alcoholic steatohepatitis is accompanied by enhanced 5-LO product levels[48], and deletion of 5-LO has (i) anti-steatotic, anti-inflammatory, and anti-fibrotic effects in experimental models of liver disease[49], (ii) protects from hepatic inflammatory injury, and (iii) improves insulin signaling at high-fat diet[14]. Based on these findings, we speculate that the reduced production of 5-LO-inhibitory LCMs promotes liver pathologies where LT biosynthesis is increased. Less understood is the role of vitamin E in the resolution of inflammation, which is driven by specialized pro-resolving mediators that are generated from eicosapentaenoic acid and docosahexaenoic acid by the concerted action of 5-, 12-, and/or 15-LO[16]. Thus, 5-LO-targeting LCMs might impede resolution, unless their local concentration at sites of inflammation is strictly regulated, which seems to be the case for α-T-13′-COOH (**4a**) that rapidly peaks and then depletes in inflamed exudates. This aspect has particular relevance for fish oil supplements where Ts, including α-T (**1a**) and γ-T (**1c**), are added as an anti-oxidant because their proposed health benefits have been ascribed to the 5-LO-dependent conversion of eicosapentaenoic acid and docosahexaenoic acid to specialized pro-resolving mediators (e.g., resolvins). Interestingly, α-T-13′-COOH (**4a**) strongly increased systemic resolvin E3 levels during the resolution phase of peritonitis, which may suggest a pro-resolving function of the LCM.

Besides interference with LT formation, inhibition of 5-LO by LCMs might also modulate immune functions through suppressing ferroptosis[50]. Vitamin E has been proposed to protect from membrane lipid peroxidation and ferroptotic cell death in glutathione peroxidase (Gpx)4-deficient T-lymphocytes, which expand and provide protection from acute lymphocytic choriomeningitis virus and *Leishmania* major parasite infections[51]. Recently, ferroptosis was ascribed to LO-dependent formation of oxidized arachidonic and adrenic phosphatidylethanolamines, and inhibition of ferroptotic death by α-T (**1a**) and TEs was demonstrated[21]. Accordingly, oral vitamin E partially reduced lipid peroxidation and oxidative stress through inhibition of 5-LO in PBMCs from hemodialysis patients[52].

In conclusion, we here provide strong evidence that the vitamin E metabolite α-T-13′-COOH (**4a**) essentially contributes to the anti-inflammatory activity of α-T (**1a**) as endogenous, high-affinity inhibitor of 5-LO. Moreover, we provide mechanistic insights into the binding mode and inhibitory mechanism of 5-LO thereby suggesting a novel allosteric mode of enzyme inhibition. Our data strongly prompt for consideration of specific vitamin E metabolism when recommending dietary doses, designing clinical studies, and evaluating the therapeutic value of vitamin E in the context of personalized medicine.

## Methods

**Materials**. α-T (**1a**; ≥96%) was purchased from Enzo Life Sciences (Loerrach, Germany), β-T (**1b**; >99%), γ-T (**1c**; ≥96%), δ-T (**1d**; ≥90%) were obtained from Sigma-Aldrich (Deisenhofen, Germany), and α- (**1e**; ≥98%), β- (**1f**; ≥98%), γ- (**1g**; ≥98%), δ-TE (**1h**; ≥98%), 3,4-dihydro-6-hydroxy-α,2,5,7,8-pentamethyl-2*H*-1-benzopyran-2-pentanoic acid (α-CMBHC, α-T-5′-COOH, **5a**; ≥98%), 2,5,7,8-tetramethyl-2-(2′-carboxyethyl)-6-hydroxychroman (α-CEHC, α-T-3′-COOH, **6a**; ≥98%), 2,7,8-trimethyl-2-(2′-carboxyethyl)-6-hydroxychroman (γ-CEHC, γ-T-3′-COOH, **6c**; ≥98%) and 2,8-dimethyl-2-(2′-carboxyethyl)-6-hydroxychroman (δ-CEHC, δ-T-3′-COOH, **6d**; ≥95%) were from Cayman Chemicals (Ann Arbor, MI). Vitamin E forms and metabolites were dissolved in DMSO, stored in the dark at −20 °C under argon, and freezing/thawing cycles were kept to a minimum. Zileuton was from Carbosynth (Berkshire, UK); sesamin was from Biopurify Phytochemicals (Chengdu, China).

**Isolation and semi-synthesis of vitamin E derivatives**. δ-TE-13′-COOH (**4h**; ≥98%) was isolated from *G. kola*, and α-T-13′-COOH (**4a**; ≥98.5%), δ-T-13′-COOH (**4d**; ≥95%), α-T-13′-CH₂OH (**2a**; ≥96%), and δ-T-13′-CH₂OH (**2d**; ≥96%) were semi-synthesized[53]. δ-TE-13′-CH₂OH (**2h**; ≥95%), γ-TE-12a′-CH₂OH (**3g**; ≥94%), δ-TE-12a′-CH₂OH (**3h**; ≥95%), δ-TE-12a′/13′-diCH₂OH (**8**; ≥99%), γ-TE-12a′/13′-diCH₂OH (**9**; ≥96%), the adduct (δ,γ)-bi-O-amplexichromanol (**10**; ≥98%), and γ-TE-13′-COOH (**4g**; ≥95%) were isolated from stem barks of *G. amplexicaulis*, an endemic tree from New Caledonia[54,55]. α-T-13′-COOH (**4e**; ≥98.5%) and β-TE-13′-CH₂OH (**4f**; ≥99%) were semi-synthesized from δ-TE-13′-COOH (**4h**)[56]. Chemical procedures for the multi-step semi-synthesis of α-TE-13′-CH₂OH (**2e**; ≥99%), β-TE-13′-CH₂OH (**2f**; ≥99%), δ-DE-13′-CH₂OH (**4l**; ≥99%), and δ-TE-13′-CONHBz (**7**; ≥99%) and the corresponding spectral data are described in Supplementary Note 1. The purity of the original semi-synthetic compounds α-TE-13′-CH₂OH (**2e**), β-TE-13′-CH₂OH (**2f**), α-TE-13′-COOH (**4e**), β-TE-13′-COOH (**4f**), δ-DE-13′-COOH (**4l**), and δ-TE-13′-CONHBz (**7**) has been determined using HPLC-ELSD analysis. The purity of all the other vitamin E derivatives mentioned in this section has been assessed by means of HPLC-DAD and -FL analysis.

**Human blood, blood cell isolation, and cell lines**. Human blood (containing 3.13% sodium citrate) and leukocyte concentrates were provided by the Institute for Transfusion Medicine of the University Hospital Jena (Germany). Human PMNL, monocytes, PBMC, and platelets were freshly isolated from leukocyte concentrates[57]. The protocols for experiments with human blood and blood cells were approved by the ethical commission of the Friedrich-Schiller-University Jena. Venous blood was collected in heparinized tubes (16 I.E. heparin/ml blood) from fasted (12 h) adult (18–65 years) male and female registered healthy volunteers, with informed consent. These subjects donated blood every 8–12 weeks, had no apparent infections, inflammatory conditions, or current allergic reactions (according to prior physical inspection by a clinician) and had not taken antibiotics or anti-inflammatory drugs for at least 10 days prior to blood collection. Leukocytes were concentrated by centrifugation of the freshly withdrawn blood (4000 × *g*/20 min/20 °C) and centrifuged on lymphocyte separation medium (LSM 1077, GE Healthcare, Freiburg, Germany). PMNL were recovered from the pellet after hypotonic lysis of erythrocytes. Platelets were obtained from the supernatant and washed with PBS pH 5.9/0.9% NaCl (1:1). Monocytes were isolated from the PBMC fraction by adherence to culture flasks (Greiner, Nuertingen, Germany) for 1.5 h at 37 °C and 5% CO₂ in RPMI 1640 medium (Sigma-Aldrich) containing FCS (Sigma-Aldrich; 5%), L-glutamine (Sigma-Aldrich; 2 mM) and penicillin/streptomycin (GE Healthcare; 100 U/ml and 100 μg/ml) (monocyte medium). The purity of the monocyte preparation was >85% as defined by forward- and side-light scatter properties and detection of the CD14 surface molecule by flow cytometry (BD FACS Calibur, Heidelberg, Germany).

Human lung adenocarcinoma epithelial A549 (ATCC, Manassas, VA) and human embryonic kidney (HEK)-293 cells (ATCC) were grown in DMEM/high glucose (4.5 g/l; GE Healthcare) containing heat-inactivated FCS (10%) and penicillin/streptomycin at 37 °C and 5% CO₂. Confluent cells were detached using trypsin/EDTA (GE Healthcare) and reseeded at 1 × 10⁴ cm². Cell lines were tested for mycoplasma. HEK-293 was reported to be a misidentified cell line, which was used because stably transfected cells were available as validated experimental cell-based model for studying the regulation of 5-LO[58]. None of the cell lines used in this study were authenticated.

**Determination of 5-LO activity**. *E. coli* Bl21 (DE3) cells were transformed with pT3–5LO plasmid to express human recombinant 5-LO, which was purified by affinity chromatography on an ATP-agarose column[59]. In brief, *E. coli* was lysed in 50 mM triethanolamine/HCl pH 8.0 plus EDTA (5 mM), soybean trypsin inhibitor (60 μg/ml), phenylmethanesulphonyl fluoride (1 mM), dithiothreitol (1 mM), and lysozyme (1 mg/ml). After sonication (3 × 15 s), the homogenate was centrifuged at 10,000 × *g* for 15 min and the supernatant again at 40,000 × *g* for 70 min at 4 °C. The supernatant was applied to an ATP-agarose column (Sigma-Aldrich), and the column was washed with (i) PBS pH 7.4, 1 mM EDTA, (ii) 50 mM phosphate buffer pH 7.4, 0.5 M NaCl, 1 mM EDTA, and (iii) 50 mM phosphate buffer pH 7.4, 1 mM EDTA. 5-LO was eluted with 50 mM phosphate buffer pH 7.4, 1 mM EDTA, 20 mM ATP.

Purified 5-LO (0.5 μg) in PBS pH 7.4 containing EDTA (1 mM) and ATP (1 mM) was pre-incubated with LCMs for 10 min at 4 °C, pre-warmed for 30 s at 37 °C and 5-LO product formation was started by addition of AA (Sigma-Aldrich; 20 μM if not indicated otherwise) plus CaCl₂ (2 mM). After 10 min at 37 °C, an equal volume of ice-cold methanol was added and formed 5-LO products (all-trans isomers of LTB₄ and 5-H(P)ETE) were extracted using Sep-Pak C18 35 cc Vac Cartridges (Waters, Milford, MA). 5-LO products were separated by RP-HPLC on a Nova-Pak C18 Radial-Pak Column (4 μm, 5 × 100 mm, Waters) under isocratic conditions (73% methanol/27% water/0.007% trifluoroacetic acid) at a flow rate of 1.2 ml/min and detected at 235 and 280 nm[59]. 5-LO product formation for vehicle control: 5 μM AA, 256 ± 68 ng; 20 μM AA, 818 ± 116 ng; 40 μM AA, 262 ± 33 ng; 80 μM AA, 208 ± 72 ng. Zileuton (IC₅₀ = 0.84 ± 0.23 μM) and BWA4C (Sigma-Aldrich; IC₅₀ = 0.03 ± 0.01 μM) were used as reference 5-LO inhibitors. To investigate whether α-T (**1a**) and δ-TE-13′-COOH (**4h**) reversibly inhibit 5-LO, the purified enzyme was pre-incubated with the test compounds at 30 and 0.1 μM for 15 min and then 10-fold diluted to a final compound concentration of 3 and 0.01 μM, respectively. AA was added after incubation for 5 min on ice. Triton X-100 (Sigma-Aldrich; 0.01%) was added to the reaction buffer to recognize nuisance inhibition of 5-LO (5-LO product formation: 360 ± 63 ng). Whether δ-TE-13′-COOH (**4h**) targets the membrane binding site of 5-LO was addressed by supplementation of phosphatidylcholine from egg york (type XVI-E; Sigma-Aldrich; 100 μg/ml; 5-LO product formation: 2.3 ± 0.3 μg) and the non-binding control phosphatidylethanolamine from bovine brain (Sigma-Aldrich; 100 μg/ml; 5-LO product formation: 212 ± 50 ng).

**Analysis of cellular 5-, 12-, and 15-LO product formation**. Monocytes (5 × 10⁶/six-well plate) were stimulated with lipopolysaccharide (LPS, Sigma-Aldrich; 1 μg/ml) for 24 h at 37 °C and 5% CO₂ in monocyte medium. Cells were washed and medium was exchanged. Freshly isolated PMNL (1 × 10⁷) and stably transfected HEK-293 (1 × 10⁶) cells were suspended in PBS pH 7.4 containing 1 mg/ml glucose and 1 mM CaCl₂. After pre-incubation with test compounds for 10 min at 37 °C, cells were treated with AA (PMNL and monocytes, 20 μM; HEK-293 cells, 3–10 μM) and/or Ca²⁺-ionophore A23187 (Sigma-Aldrich; 2.5 μM) for 10 min at 37 °C (PMNL and HEK-293 cells) or 30 min at 37 °C and 5% CO₂ (monocytes). The reaction was stopped with an equal volume of methanol (PMNL and HEK-293 cells) or on ice (monocytes). For PMNL and HEK-293 cells, major 5-LO products (LTB₄, its all-trans isomers and 5-H(P)ETE), 12-HETE (12-LO product from platelets in PMNL preparations), and 15-HETE (15-LO product) were extracted and analyzed by RP-HPLC as described for 5-LO products. Cysteinyl-LTs and oxidation products of LTB₄ were not determined. A23187- and A23187/AA-treated PMNL and A23187/AA-treated HEK-293 (5-LO_wt), HEK-293 (5-LO_3W) and HEK-293 (5-LO_Arg101Asp) cells produced 135 ± 21, 636 ± 112, 300 ± 59, 133 ± 29, 55 ± 9 ng 5-LO products, respectively. PMNL synthesized 29 ± 6 ng 12-HETE and 97 ± 23 ng 15-HETE when challenged with A23187/AA. 5-LO products from monocytes were analyzed by UPLC-MS/MS. The 5-LO reference inhibitors zileuton (3 μM) and BWA4C (0.5 μM) suppressed 5-LO product formation by 78–98%.

**Immobilization of δ-TE-13′-COOH (4h) and pull-down of 5-LO**. To covalently link δ-TE-13′-COOH (**4h**) to Toyopearl AF-Amino-650 M beads (Tosoh Bioscience, Stuttgart, Germany)[60], δ-TE-13′-COOH (**4h**; 30 mg) or 4-phenoxy butyric acid (12 mg) as control were dissolved in methanol/water (98/2) at pH 5 and coupled to washed Toyopearl resin (500 μl) by addition of 1-ethyl-3-(3-dimethylaminopropyl)carbodiimide (40 mg). After 48 h at room temperature under pH control, the resin was washed and stored in methanol/water (20/80).

For pull-down of 5-LO, the drained resin (25 μl) was blocked with binding buffer (50 mM HEPES pH 7.4, 200 mM NaCl, 1 mM EDTA) containing 0.2 mg/ml milk powder and 0.01% Triton X-100 for 1 h at 4 °C. Purified human recombinant 5-LO (5 μg) was added and the mixture was incubated for 1 h. After repeated washing with binding buffer, 5-LO was eluted with 2× Laemmli buffer, heated for 5 min at 95 °C and subjected to SDS-PAGE and Western blotting. Competitive binding of δ-TE-13′-COOH (**4h**) and either AA or α-T (**1a**) to 5-LO was investigated using an excess of AA and α-T (**1a**) in pull-down buffer (100 μM each).

**SDS-PAGE and western blotting**. Equal aliquots of pull-downs were resolved on 10% SDS-PAGE gels, proteins were transferred onto Hybond ECL nitrocellulose membranes (GE Healthcare), and 5-LO was detected using an Odyssey infrared

imager (LI-COR Biosciences)[58]. The membranes were incubated with primary mouse anti-5-LO antibody (clone 6A12, 1:1000), which was provided by Dr. Dieter Steinhilber (University of Frankfurt, Frankfurt, Germany) and produced in-house[61]. Immunoreactive bands were stained with IRDye 800CW Goat anti-Mouse IgG (H + L; 1:10,000; 926–3221, LI-COR Biotechnology, Cambridge, UK). Data from densitometric analysis were background-corrected. Uncropped blots are shown in Supplementary Fig. 8.

**Pocket finding**. The search for alternative binding pockets was performed with MOE's pocket finder (Chemical Computing Group, Montreal, Canada; https://www.chemcomp.com/journal/sitefind.htm). Cavities that exceeded a volume of 110 Å$^3$ were investigated.

**Molecular docking**. The molecular docking simulation was conducted with GOLD 5.2 (CCDC, Cambridge, UK). Scoring was calculated using the CHEMPLP (https://www.ccdc.cam.ac.uk/solutions/csd-discovery/components/gold/?caseid=5d1a2fc0-c93a-49c3-a8e2-f95c472dcff0) scoring function. The protein target structure was based on stable 5-LO (pdb entry 3o8y, https://www.ncbi.nlm.nih.gov/protein/3O8Y_A). Four virtual mutations were inserted to return to the 5-LO wild-type sequence: E13W, H14F, G75W and S76L. The residues were exchanged and the structure was energetically minimized in Discovery Studio (version 3.5; Biovia, San Diego, CA). Hydrogens were added with GOLD, using default settings. The binding site was defined in an 8 Å radius around C2 of residue I167. Protein–ligand interactions of the docking poses were analyzed using LigandScout 3.12 (www.inteligand.com/ligandscout).

**Stable expression of mutant 5-LO in HEK-293 cells**. To stably express 5-LO and its mutants, HEK-293 cells were transfected with pcDNA3.1/neom (+)_5-LO and respective 5-LO mutants using Lipofectamine® LTX (Thermo Fisher Scientific, Waltham, MA)[58]. The pcDN3.1/neom (+) vectors for 5-LO (wt) and 3W mutant expression were generous gifts by Dr. Dieter Steinhilber. The 5-LO Arg101Asp coding sequence from the E. coli expression vector pt3_5-LO_Arg101Asp (provided by Dr. Olof Rådmark, Karolinska Institutet, Stockholm, Sweden) was cloned into the vector pcDNA3.1/neom_(+)_5-LO_Arg101Asp. EcoRI and XbaI restriction sites were introduced by polymerase chain reaction. The sequences of 5-LO (wt) and the site-directed mutants in the expression constructs were confirmed by sequencing by Eurofins Genomics (Ebersberg, Germany) and are shown in Supplementary Table 5. Reverse primer used was 5′-GTTTTTGCCGTGTTTCCAGT-3′ as listed in Supplementary Table 6. Cells were incubated for 48 h at 37 °C and 5% CO$_2$ and subsequently selected by geneticin (400 μg/ml) for 2–4 weeks.

**Animals**. Male CD-1 mice (33–39 g, 6–8 weeks, Charles River Laboratories, Calco, Italy) or female BALB/c mice (8 weeks, Charles River Laboratories) were housed in a controlled environment (21 ± 2 °C) and provided with standard rodent chow and water. Animals were allowed to acclimate for four days prior to experiments and were subjected to 12 h light/12 h dark schedule. Mice were randomly assigned for the experiments, which were conducted during the light phase. The experimental procedures, according to Italian (DL 26/2014) and European (n. 63/2010/UE) regulations on the protection of animals used for experimental and other scientific purposes, were approved by the Italian Ministry (Number 584/2015-PR; Number 1092/2015-PR).

**Zymosan-induced peritonitis in mice**. α-T-13′-COOH (**4a**), δ-TE-13′-COOH (**4h**), α-T (**1a**), zileuton or vehicle (10 mg/kg, i.p., 2% DMSO; 0.5 ml) were administered to CD-1 mice 30 min before injection of zymosan (Sigma-Aldrich; 2 mg/ml in saline, 0.5 ml, i.p.). Alternatively, α-T (**1a**; 50 mg/kg, p.o.), zileuton (30 mg/kg, p.o.) or vehicle (0.5 ml of 0.5% carboxymethylcellulose and 10% Tween 20) were given 1 h prior zymosan injection, in combination with sesamin (250 mg/kg, p.o.), as indicated. Mice were killed by inhalation of CO$_2$ after another 30 min for determination of LTC$_4$ levels and vascular permeability and after 4 h for the analysis of LTB$_4$ levels, lipidomic profiles and immune cell infiltration[35]. Cell infiltration and lipid mediator profiles during resolution were studied after 18 h. Peritoneal exudates were collected and cells counted after vital trypan blue staining. LTC$_4$, LTB$_4$, and PGE$_2$ were analyzed in the exudate by enzyme immunoassays (Enzo Life Sciences, Lörrach, Germany) according to manufacturer's instructions. To assess vascular permeability, Evans blue dye (40 mg/kg in saline, 0.3 ml; Sigma-Aldrich) was injected into the tail vein immediately before the induction of peritonitis[35]. Exudates were centrifuged (3000 × g, 5 min) and the absorbance of the supernatants was measured at 610 nm using a Beckman Coulter DU730 spectrophotometer. To investigate the effect of α-T (**1a**) on the accumulation of α-T-13′-COOH (**4a**) in infiltrated leukocytes after peritonitis has been established, vehicle or α-T (**1a**; 50 mg/kg, p.o.) were given 1 h before and/or 4 h post zymosan injection, and mice were killed after another 1.5 h. Leukocytes were obtained from peritoneal exudates by centrifugation (1200 × g, 10 min, 4 °C).

**Airway responsiveness measurements**. BALB/c mice were sensitized to OVA, and bronchial reactivity and pulmonary LTC$_4$ levels were measured[62]. On days 0 and 7, α-T-13′-COOH (**4a**; 10 mg/kg, i.p.) or vehicle (4% DMSO in saline; 0.5 ml)

were administered 30 min before injection of OVA (100 μg; Sigma Aldrich) which was adsorbed to aluminum hydroxide gel (3.3 mg; Sigma Aldrich). Mice were sacrificed on day 21, and bronchi were collected. Bronchial rings of 1–2 mm length were cut and placed in organ baths mounted to isometric force transducers (Type 7006, AD Instruments, Ugo Basile, Comerio, Italy) and connected to a Powerlab 800 (AD Instruments). Rings were initially stretched until a resting tension of 0.5 g was reached and allowed to equilibrate for at least 30 min. In each experiment, bronchial rings were challenged with carbachol ($10^{-6}$ M) until the response was reproducible. Bronchial reactivity was then assessed from a cumulative concentration–response curve to carbachol ($1 \times 10^{-8} - 3 \times 10^{-5}$ M). To determine pulmonary LTC$_4$ levels, lungs were isolated and homogenized in PBS pH 7.4 (100 mg tissue/ml) prior to detection of LTC$_4$ by enzyme immunoassay (Cayman Chemicals).

**Multi-organ-tissue-flow (MOTiF) biochips**. Biochips were made by injection molding from polystyrene (PS)[63,64]. A 12 μm thick polyethylene terephthalate (PET) membrane (TRAKETCH) with a pore diameter of 8 μm and a pore density of $1 \times 10^5$ pores/cm² (Sabeu, Radeberg, Germany) was integrated in the upper and lower part of the biochip by heat-sealing with the bulk material (Supplementary Fig. 4a, b). Chips and channel structures were sealed on the top and bottom sides with an extruded 125 μm thick PS bonding foil using a low temperature bonding method. Upper and the lower parts of the biochips were assembled by a double-sided adhesive film. Oxygen plasma treatment for hydrophilization of the whole chip surface was performed to support cell adhesion and to reduce air bubble formation in the chips.

HepaRG cells (Biopredic International, Rennes, France) were cultivated at 37 °C and 5% CO$_2$ in William's Medium E (Biochrom, Berlin, Germany) containing 10% FCS (GIBCO, Darmstadt, Germany), 5 μg/ml insulin (Sigma-Aldrich), 2 mM glutamine (GIBCO), 0.5 μM hydrocortisone-hemisuccinate (Sigma-Aldrich), 100 U/ml penicillin (GIBCO) and 100 μg/ml streptomycin (GIBCO) (hepatocyte growth medium) for 14 days before differentiation. Medium was renewed every 3–4 days. HepaRG differentiation was induced by medium supplementation with 2% DMSO for 14 days. Differentiated cells were used up to 4 weeks.

HUVECs were isolated from human umbilical cord veins (HUVEC)[65]. Experiments were performed with HUVECs cultured in Endothelial Cell Medium (ECM) (Promocell, Heidelberg, Germany) up to passage 4. HUVECs were seeded at a density of $1.5 \times 10^5$ cells/cm² and cultured up to 95% confluence before being sub-cultured.

LX-2 stellate cells were kindly provided by Dr. Ralf Weiskirchen (RWTH University Hospital Aachen, Germany). LX-2 cells were cultured in Dulbecco's Minimum Essential Medium (DMEM) (Biochrom) supplemented with 10% FCS, 1 mM sodium pyruvate (GIBCO) and penicillin/streptomycin and seeded at a density of $2.0 \times 10^5$ cells/cm². Cells were grown up to 95% confluence before being sub-cultured.

PBMCs of three different healthy donors were isolated by Biocoll (Biochrom) density gradient centrifugation and seeded in 6-well dishes with a density of $1.0 \times 10^7$ cells/cm² in X-VIVO 15 medium (Lonza, Cologne, Germany) supplemented with 10% autologous human serum, 10 ng/ml human granulocyte macrophage colony-stimulating factor (GM-CSF) (PeproTech, Hamburg, Germany) and penicillin/streptomycin. After 1 h incubation in a humidified cell incubator at 5% CO$_2$ and 37 °C, the cells were washed twice with X-VIVO 15 medium and subsequently used for liver organoid assembly. Typically, $1 \times 10^5$ monocytes per ml whole blood were obtained. Monocytes were harvested with 4 mg/ml lidocain (Sigma-Aldrich) supplemented with 5 mM EDTA (Sigma-Aldrich).

For the preparation of "hepatocyte only biochips", $6 \times 10^5$ differentiated HepaRG cells were seeded at the bottom of the upper membrane and on top of the lower membrane in hepatocyte growth medium[66]. Liver models were assembled by staggered seeding of vascular and hepatic cell layers. In each sterilized biochip, $3 \times 10^5$ HUVECs and $1 \times 10^5$ monocytes were mixed and seeded on top of the upper membrane in the upper chamber and pre-incubated 3–4 h to allow attachment. Subsequently, biochips were flipped upside down, and $2 \times 10^5$ HUVECs and $6.7 \times 10^4$ monocytes were mixed and seeded on the bottom of the membrane in the lower chamber.

HUVEC/monocytes were co-cultured for at least 5 days with a daily medium exchange in ECM for the first two days and after reaching confluence in M199 medium (GIBCO) supplemented with 10 ng/ml epidermal growth factor (Sigma-Aldrich), 90 μg/ml heparin (Sigma-Aldrich), 2.8 μM hydrocortisone, 7.5 μg/ml endothelial mitogen (Merck Chemicals Darmstadt, Germany), 5 μg/ml ascorbic acid (Sigma-Aldrich), 10 ng/ml GM-CSF to induce macrophage differentiation, 100 U/ml penicillin, 100 μg/ml streptomycin, and 10% autologous human serum from monocyte donors. Subsequently, $1 \times 10^6$ differentiated HepaRG cells and $3.3 \times 10^4$ LX-2 cells were seeded at the bottom of the upper membrane, cultured for 24 h and equally seeded on top of the lower membrane in hepatocyte growth medium two to four days prior to experimental use. Medium 199 supplemented with 10% FCS, 680 mM L-glutamine, 25 μg/ml heparin, 7.5 μg/ml endothelial mitogen, 5 μg/ml vitamin C, 10% autologous human serum, 100 U/ml penicillin, and 100 μg/ml streptomycin was used in the upper and lower biochip chamber and hepatocyte growth medium in the medium biochip chamber.

After rinsing the chambers of the biochip with medium as indicated above, vehicle (DMSO), α-T (**1a**; 100 μM) or sesamin (3 μM) were added. Biochips were

incubated for the indicated time, medium was collected, and the system was washed with methanol (500 µl). Medium and methanol fractions were combined, and LCMs were extracted and analyzed by UPLC-MS/MS.

**Treatment of blood and cells for lipid mediator profiling.** LPS-activated monocytes were treated with A23187 (2.5 µM) and AA (20 µM) to induce lipid mediator biosynthesis as described for LO product formation. Human blood was pre-incubated with vehicle (DMSO) or α-T-13′-COOH (**4a**, 100 nM) for 10 min, primed with LPS (1 µg/ml) for 30 min and stimulated with N-formyl-methionyl-leucyl-phenylalanine (fMLP, Sigma-Aldrich; 1 µM) for 15 min at 37 °C.

**Extraction of lipid mediators.** Lipid mediators were extracted from pelleted cells (800 × g, 10 min, 4 °C), human blood (2 ml), blood plasma (0.3 ml) and peritoneal exudates (1 ml) using Sep-Pak C18 35 cc Vac Cartridges (Waters)[67]. Aqueous samples were diluted in PBS pH 7.4 (530 µl), acidified with 1 M HCl (30 µl), supplemented with the internal standard PGB$_1$ (100 ng, Cayman Chemicals) and added to the columns. After washing of the columns with water twice (0.5 ml, each), lipid mediators were eluted with methanol (300 µl) and analyzed by UPLC-MS/MS.

To extract lipid mediators of the resolution phase of peritonitis from mouse plasma and peritoneal exudates[17], aliquots were combined with ice-cold methanol (2 ml) and d8-5S-HETE, d4-LTB$_4$, d5-lipoxin A$_4$, d5-resolvin D$_2$, d4-PGE$_2$ (200 nM, each, Cayman Chemicals) and d8-AA (10 µM, Cayman Chemicals) were added as internal standards. Samples were incubated at −20 °C for 1 h to precipitate proteins and then centrifuged (1200 × g, 10 min, 4 °C). The supernatant was transferred to acidified water (8 ml) to adjust pH = 3.5 and loaded onto solid phase cartridges (Sep-Pak® Vac 6cc 500 mg/6 ml C18; Waters), which were equilibrated with methanol (6 ml) and water (2 ml). Columns were washed with water (6 ml) and hexane (6 ml), and lipid mediators were eluted with methyl formiate (6 ml). Eluates were evaporated to dryness and dissolved in methanol/water (50/50) for UPLC-MS/MS analysis.

**Extraction of LCMs.** Lipids, including LCMs, were extracted from PMNL, murine peritoneal exudates or murine plasma by successive addition of PBS pH 7.4, methanol, chloroform and saline (final ratio: 14:34:35:17)[68]. After evaporation of the organic layer, extracted LCMs were dissolved in 100 µl methanol and diluted. δ-TE-13′-COOH (**4h**; 0.6 nmol) was added as internal standard.

**Reversed phase liquid chromatography and mass spectrometry.** Eicosanoids, docosanoids and free fatty acids were separated on an Acquity UPLC BEH C18 column (1.7 µm, 2.1 × 50 mm, Waters) using an Acquity™ UPLC system (Waters) and detected by multiple reaction monitoring using a QTRAP 5500 Mass Spectrometer (Sciex, Darmstadt, Germany) equipped with an electrospray ionization source[17,67]. Chromatography was conducted at a flow rate of 0.8 ml/min and a column temperature of 45 °C using acetonitrile plus 0.07% formic acid as mobile phase A and water/acetonitrile (90/10) plus 0.07% formic acid as mobile phase B. After isocratic separation at A/B = 30/70 for 2 min, a linear gradient to A/B = 70/30 was applied within 5 min. Multiple reaction monitoring in the negative ion mode was used to detect diagnostic ion fragments[67]. Data were normalized on the internal standard PGB$_1$ and are given as relative intensities. The analytic method was optimized to compare lipid mediator profiles between samples and not for absolute quantification.

For the resolution phase of peritonitis, lipid mediators were separated on an Acquity UPLC BEH C18 column (1.7 µm, 2.1 × 100 mm, Waters) at 0.3 ml/min and 50 °C. The mobile phase was acidified with 0.01% acetic acid and consisted of methanol/water (42/58) that was ramped to methanol/water (86/14) over 12.5 min followed by isocratic elution with methanol/water (98/2) for 3 min. Diagnostic ion fragments of lipid mediators were detected in the negative ion mode using scheduled multiple reaction monitoring (window: 60 s)[17]. The amounts of lipid mediators and fatty acids were calculated based on 6 internal and 45 external standards[17].

LCMs were separated on an Acquity UPLC BEH C18 column (1.7 µm, 2.1 × 50 mm, Waters) at a flow rate of 0.8 ml/min at 45 °C: 0 to 4.5 min, gradient from 50% mobile phase A (acetonitrile/water, 10/90, 0.07% formic acid)/50% mobile phase B (acetonitrile, 0.07% formic acid) to 100% B; 4.5 to 5.5 min, isocratic at 100% B. Multiple reaction monitoring was used to detect LCMs in the negative ion mode. The precursor-to-product ion transitions were selected according to Jiang et al.[69] and are listed in Supplementary Table 3. The ion spray voltage was set to −4500 V, the heated capillary temperature to 450 °C, the sheath gas pressure to 50 psi, the auxiliary gas pressure to 40 psi, and the declustering potential to 80 eV (for Ts) or 110 eV (for TEs). Collision energies and lower limits of quantitation are listed in Supplementary Table 3. After normalizing the signal intensities of LCMs to the internal standard δ-TE-13′-COOH (**4h**), LCM concentrations were calculated from external calibration curves as specified in Supplementary Table 3.

Automatic peak integration was performed with Analyst 1.6 software (Sciex) using IntelliQuan default settings.

**Cell viability assay.** Freshly isolated human PBMC were incubated with vitamin E derivatives for 24 h at 37 °C and 5% CO$_2$, and cell viability was determined using a MTT assay[67]. PBMC in monocyte medium were incubated with MTT (5 mg/ml, 20 µl; Sigma-Aldrich) in darkness for 4 h at 37 °C and 5% CO$_2$. The formazan product was solubilized with SDS (10% in 20 mM HCl), and absorbance was measured at 570 nm using a Multiskan Spectrum microplate reader (Thermo Fisher Scientific). Formazan formation was inhibited by staurosporine (Merck Chemicals; 3 µM; 71.4 ± 6.2% inhibition) and abolished by lysis of cells with Triton X-100 (1%).

**Determination of radical scavenging activity.** Compounds were incubated with 2,2-diphenyl-1-picrylhydrazyl (DPPH, Sigma-Aldrich; 50 mM) in ethanol for 30 min. Absorbance was measured at 520 nm using a Multiskan Spectrum Microplate Reader (Thermo Fisher Scientific)[35]. Ascorbic acid (Sigma-Aldrich; 50 µM), used as control, scavenged DPPH radicals by 82.4 ± 1.9%.

**Measurement of intracellular Ca$^{2+}$ levels.** PMNL (1 × 10$^6$) were incubated with Fura-2/AM (Thermo Fisher Scientific; 2 nmol) for 45 min at 37 °C, washed, resuspended in PBS pH 7.4 plus 1 mg/ml glucose and transferred to fluorescene 96-well microplates (Greiner). After pre-incubation with compounds for 10 min, 1 mM Ca$^{2+}$ was added, and cells were stimulated with fMLP (1 µM) for another 10 min. Intracellular Ca$^{2+}$ concentrations were determined using a thermally (37 °C) controlled NOVOstar fluorescence microplate reader [BMG Labtech, Ortenberg, Germany; emission at 510 nm, excitation at 340 nm (Ca$^{2+}$-bound Fura-2) and 380 nm (free Fura-2)][17]. The maximal fluorescence signal was recorded after lysis of PBMC with Triton X-100, and the minimal fluorescence was assessed by adding 20 mM EDTA to chelate Ca$^{2+}$. The difference in the signal ratio obtained with Triton X-100 and Triton X-100 plus 20 mM EDTA was set to 100% for each experiment.

**Determination of 5-LO/FLAP interactions.** PBMC were seeded onto glass coverslips in growth medium for 1.5 h (37 °C, 5% CO$_2$). Cells were washed and pre-incubated with 3 µM α-T-13′-COOH (**4a**), δ-TE-13′-COOH (**4h**) or vehicle control (0.1% DMSO) in PGC buffer (PBS pH 7.4, 0.1% glucose, 1 mM CaCl$_2$) for 15 min at 37 °C prior to stimulation with A23187 (2.5 µM, 10 min, 37 °C). To analyze in situ protein interaction of 5-LO with FLAP by proximity ligation assay[17], the incubation was stopped by fixation with paraformaldehyde (4%, 20 min, room temperature). Cells were permeabilized using 100% ice-cold acetone (5 min, 4 °C) and incubated with primary antibodies against 5-LO (clone 6A12, 1:100)[61] and FLAP (polyclonal rabbit anti-FLAP antibody, 5 µg/ml, ab85227, Abcam, Cambridge, UK)[17] for 16 h at 4 °C and then treated for 1 h at 37 °C with oligonucleotide-labeled specific secondary antibodies (Duolink In Situ PLA probe anti-mouse MINUS, 1:5, DUO92004 and anti-rabbit PLUS, 1:5, DUO92002, Sigma-Aldrich). Two other circle-forming DNA oligonucleotides and a ligase were added to create a DNA circle from the antibody-bound oligonucleotides, which is possible when 5-LO and FLAP are <40 nm distant from each other. After rolling-circle-amplification (90 min, 37 °C), the newly generated DNA was visualized by hybridization with fluorescently labeled oligonucleotides (Duolink In Situ Detection Reagents FarRed, DUO92013, Sigma-Aldrich). Nuclear DNA was stained with DAPI-containing ProLong diamond antifade mountant (Invitrogen, Darmstadt, Germany). Cells were visualized by a Zeiss Axiovert 200M microscope and a Plan Neofluar ×100/1.30 Oil (DIC III) objective (Carl Zeiss, Jena, Germany), and image acquisition was performed using an AxioCam MR camera (Carl Zeiss). The interaction between 5-LO and FLAP appears as fluorescent spot (magenta). Overview images were obtained using a Plan Neofluar 40/1.30 Oil (DIC III) objective (Carl Zeiss).

**Uptake of vitamin E metabolites by PMNL.** Freshly isolated PMNL (1 × 10$^7$/ml) were suspended in PBS pH 7.4 containing 1 mg/ml glucose. After incubation with vehicle (DMSO, 0.1%), α-T-13′-COOH (**4a**) or δ-TE-13′-COOH (**4h**; 150 nM, each) for 20 min at 37 °C, cells were centrifuged (1200 × g, 5 min, 4 °C) and washed three times with PBS pH 7.4 plus 0.1% fatty acid-free BSA (Sigma-Aldrich). LCMs were extracted and analyzed by UPLC-MS/MS using γ-TE-13′-COOH (**4g**; 12.5 pmol) as internal standard. Intracellular vitamin E metabolite concentrations were calculated assuming spherical cell shape, an equal intracellular distribution, and an average diameter of 13 µm (as determined using a Vi-CELL Series Cell Counter, Beckman Coulter, Krefeld, Germany).

**Determination of LTC$_4$ synthase activity.** To investigate the effect of LCMs on LTC$_4$S activity, we used microsomal membranes from HEK-293 cells stably expressing LTC$_4$S[70]. Membranes (2.5 µg total protein) were treated with LCMs for 10 min at 4 °C in potassium phosphate buffer (0.1 M, pH 7.4) containing 5 mM glutathione. The conversion of LTA$_4$ to LTC$_4$ methyl esters was initiated by addition of LTA$_4$ methyl ester (Cayman Chemicals; 1 µM). After 10 min at 4 °C, the reaction was terminated by ice-cold methanol (one volume), and LTC-d5 methyl ester (5 ng; Cayman Chemicals) was added as internal standard. LTC$_4$ was extracted and analyzed by UPLC-MS/MS as described for lipid mediators. Vehicle (DMSO)-treated microsomal membranes produced 7.1 ± 0.5 nmol LTC$_4$. The LTC$_4$ reference inhibitor MK-886 (Cayman Chemicals, 10 µM) suppressed LTC$_4$ formation by 95.6 ± 0.9%.

**Activity assays of isolated COX-1 and -2.** Purified bovine COX-1 (Cayman Chemicals; 50 units) or human recombinant COX-2 (Cayman Chemicals; 20 units) in 100 mM Tris buffer pH 8, 5 mM glutathione, 5 μM hemoglobin, and 100 μM EDTA were pre-incubated with LCMs for 5 min at 4 °C followed by 1 min at 37 °C. Formation of COX-derived 12(S)-hydroxy-5-cis-8,10-trans-heptadecatrienoic acid (12-HHT) was initiated by addition of AA (COX-1: 5 μM; COX-2: 2 μM). 12-HHT was extracted using Sep-Pak C18 35 cc Vac Cartridges (Waters) and separated by RP-HPLC on a Nova-Pak C18 Radial-Pak Column (4 μm, 5 × 100 mm, Waters) as described for 5-LO products. The COX-1/2 reference inhibitor indomethacin (Sigma-Aldrich; 10 μM) suppressed COX-1 product formation by 79.3 ± 2.3% and COX-2 product formation by 77.4 ± 3.1%.

**Determination of mPGES-1 activity.** Microsomal membranes of interleukin-1β-treated A549 cells were prepared as source of mPGES-1[67]. In brief, cells were treated with interleukin-1β (2 ng/ml) at 37 °C and 5% $CO_2$ for 48 h, harvested and frozen in liquid nitrogen. Ice-cold homogenization buffer (0.1 M potassium phosphate buffer pH 7.4, 1 mM phenylmethanesulphonyl fluoride, 60 μg/ml soybean trypsin inhibitor, 1 μg/ml leupeptin, 2.5 mM glutathione, and 250 mM sucrose) was added, and cells were incubated for 15 min on ice prior to sonication (3 × 20 s, 4 °C). The microsomal fraction was obtained by differential centrifugation at 10,000 × g for 10 min and 174,000 × g for 1 h at 4 °C. Microsomes were suspended in homogenization buffer and diluted in potassium phosphate buffer (0.1 M, pH 7.4) containing 2.5 mM glutathione. Membranes (2.5–5 μg total protein) were pre-incubated with LCMs for 15 min at 4 °C, and PGE2 formation was initiated by addition of $PGH_2$ (20 μM, final concentration). The reaction was terminated after 1 min by addition of one volume stop solution (40 mM $FeCl_2$, 80 mM citric acid and 1 nmol of the internal standard 11β-PGE2, Cayman Chemicals). PGE2 was extracted using Sep-Pak C18 35 cc Vac Cartridges (Waters), separated by RP-HPLC on a Nova-Pak C18 Radial-Pak Column (4 μm, 5 × 100 mm, Waters) under isocratic conditions (30% acetonitrile/70% water/ 0.007% trifluoroacetic acid) at a flow rate of 1 ml/min and detected at 195 nm[67]. The mPGES-1 reference inhibitor MK-886 (10 μM) suppressed PGE2 formation by 78.0 ± 2.5%.

**Determination of COX-1 activity in washed platelets.** Freshly isolated human platelets (1 × 10^8) were pre-incubated with LCMs for 5 min at room temperature. Formation of COX-1-derived 12-HHT was initiated by addition of exogenous AA (5 μM). After 5 min at 37 °C, 12-HHT was extracted using Sep-Pak C18 35 cc Vac Cartridges (Waters) and separated by RP-HPLC on a Nova-Pak C18 Radial-Pak Column (4 μm, 5 × 100 mm, Waters) as described for 5-LO products.

**Determination of sEH activity.** To investigate the effect of LCMs on sEH activity, we purified the human recombinant enzyme from sEH expressing Sf9 insect cells by benzylthio-sepharose affinity chromatography[70]. Sf9 cells were infected with a recombinant baculovirus (72 h) and sonicated (3 × 10 s, 4 °C) in lysis buffer (50 mM $NaHPO_4$ pH 8, 300 mM NaCl, 10% glycerol, 1 mM EDTA, 1 mM phenylmethansulfonyl fluoride, 10 μg/ml leupeptin, 60 μg/ml soybean trypsin inhibitor). Supernatants were centrifuged (100,000 × g, 60 min, 4 °C) and subjected to benzylthio-sepharose affinity chromatography. sEH was eluted with 4-fluorochalcone oxide in PBS pH 7.4, 1 mM dithiothreitol and 1 mM EDTA and then dialyzed and concentrated. The purified enzyme (60 ng) was diluted in 25 mM Tris HCl, pH 7 plus 0.1 mg/ml BSA and pre-incubated with LCMs for 10 min at room temperature. The enzymatic reaction was started by addition of PHOME (50 μM, Cayman Chemicals), which was converted to the fluorescent 6-methoxy-naphtaldehyde. After 60 min in darkness, the reaction was terminated by $ZnSO_4$ (200 mM), and the fluorescence was measured using a NOVOstar fluorescence microplate reader (BMG Labtech, excitation: 330 nm, emission: 465 nm). The reference inhibitor AUDA (Cayman Chemicals; 100 nM) suppressed sEH activity by 42.2 ± 1.9%.

**Statistics.** Data are presented as mean ± s.e.m. of n observations, where n represents the number of experiments or the number of animals, as indicated. Sample size was not pre-determined by statistical methods. Data from animal studies were confirmed as normally distributed with similar variance between groups using a Kolmogorov–Smirnov test. Samples of the vitamin E library were blinded for screening purposes but not for further biological evaluation, and samples from mouse peritoneum were blinded for counting infiltrated cells. Outliers were determined using a Grubb's test. Comparison of different groups was performed by one-way or two-way ANOVA for independent or correlated samples followed by Tukey or Bonferroni HSD post hoc test or by two-tailed Student t test for paired or unpaired samples. Tests were conducted using a two-sided α level of 0.05. P values < 0.05 were considered statistically significant. Statistical calculations were performed using GraphPad Prism 4.0 (GraphPad Software, La Jolla, CA). $IC_{50}$ values were determined by graphical analysis using SigmaPlot 13.0 (Systat Software Inc., San Jose, CA).

## Data availability
The data that support the findings of this study are available within the article and supplementary files or from the corresponding authors upon request.

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

## Acknowledgements

The authors are grateful to the North and South Provinces of New Caledonia who facilitated our field investigation, as well as to Marc Litaudon, Vincent Dumontet and Cyril Poulain for *G. amplexicaulis* sample collection. Moreover, the authors thank Katrin Fischer, Verena Krauth, Felix Nikels, Stefanie Liening, Monika Listing, Marius Melzer, Erik Romp, Olga Sailer, Bärbel Schmalwasser, Heidi Traber, and Petra Wiecha for technical assistance in performing experimental methods. A.K. was supported by the German Research Council (GRK 1715, KO 4589/7-1), a research initiative for young scientists of the University of Jena (DRM/2015-01), and the German Academic Exchange Service (57389523) paid from the German Federal Ministry of Education and Research. U.G. was supported by the German Research Council (GA 2101/2-1). O.W. was funded by the Free State of Thuringia and the European Social Fund (2016 FGR 0045), the Free State of Thuringa, ProExcellence Initiative 2 ("RegenerAging"), and the German Research Council (SFB1127 ChemBioSys and SFB1278 Polytarget). Work of S.L. was supported by the German Research Council (RTG 1715), by the Free State of Thuringia and the European Social Fund (2016 FGR 0045), the Federal Ministry of Education and Research (01EA1411A), and the German Ministry of Economics and Technology Energy (AiF 16642 BR) via AiF (German Federation of Industrial Research Associations) and FEI (Research Association of the German Food Industry). K.A. received a Ph.D. grant from the Government of Syria. A.V. and C.-P.D. received Ph.D. grants from the University of Angers. D.S. is an Ingeborg Hochmair Professor at the University of Innsbruck. The authors further thank Inte:Ligand GmbH (Vienna, Austria) for an academic license of LigandScout.

## Author contributions

B.W., D.G., D. Séraphin, H.S., J.H., M.B., M.W., P.R., and S.L. isolated and provided natural products. A.V., C.D., D. Séraphin, G.V., J.H., K.A., M.B., and P.R. semi-synthesized derivatives. S.P., A.K., A.R., and L.S. designed and S.P, A.R., F.T., F.R., R.B., and K.N. conducted and analyzed animal studies. A.K. designed, and H.P. and S.K.

performed and analyzed cell-free and cell-based experiments. V.T. and D. Schuster performed molecular docking simulations. A.M. provided the liver-on-chip and H.P. and M.R. performed the experiments. H.P. and A.K. quantified LCMs in biological samples. U.G. performed mutagenesis studies and the proximity ligation assay. C.W. and S.R. provided leukocyte concentrates. A.K., A.M., A.R., D. Séraphin, D. Schuster, F.R., H.P., J. H., M.R., L.S., P.R., S.P., O.W., V.T. interpreted the experiments. A.K. conceived the project and wrote the manuscript, which was edited and approved by all authors.

## Additional information

**Competing interests:** A.K., B.W., D.G., D. Schuster, D. Séraphin, H.S., J.-J.H., P.R., O.W., and V.T. declare competing financial interest. They are inventors of a patent related to the development of LCM-inspired 5-LO inhibitors. The remaining authors declare no competing interests.

