## [Peer Review File · Nature Communications]

Reviewers' comments:

Reviewer #1 (Remarks to the Author):

This report demonstrates that metabolites of alpha-tocopherol inhibit 5-LO. The biochemistry analysis for these interactions is outstanding and statistics are appropriate. Moreover, this is a very important contribution to the field. However there are some comments.

1. Page 4 line 5-6. What forms of vitamin E? Indicate the isoforms when discussing studies in the introduction rather than using the term vitamin E. Vitamin E is used to incorrectly indicate alpha-tocopherol when vitamin E really is a general term encompassing all the isoforms.
2. Page 4 line 15. Delete the word "Thus". This sentence is not a conclusion of the previous sentence. It makes it confusing.
3. Page 7 line 6. Indicate where in the supplementary information because it is at the end and hard to find. Indicate something like "supplementary note in supplement information"
4. Fig 1e should have a more thorough description of the length of times for first concentration and diluted concentration and when substrate was present.
5. For Fig 1f, as a control to demonstrate specificity, include competition with increasing concentrations of non-labeled compounds.
6. In the legend of Fig 1, why is the 4-phenoxy butyric acid the control? This is confusing. Provide rationale.
7. Page 9 line 10. d-TE-13'-COOH compound is not in Fig. 1b but DE-13'-COOH is in the figure. Add d-TE-13'-COOH and its IC50 to Fig 1b.
8. Page 9 line 11. Nuisance is vague. Does this mean nonspecific? background?....
9. Page 9 line 27. What is meant by "retain" in this sentence?
10. Page 10 text. For clarity, use the same AA nomenclature in the text as that in the figure 1i.
11. Figure 2. panel e and f are confusing as to how they are different than panel e and d. It needs to be more clearly labeled in the figure that e and f have a-T in both groups per panel.
12. Page 13 line 25. In the methods aT is given either i.p. or orally. In the figure legends it is not clearly indicated whether the tocopherol in each panel is either i.p. or orally administered.
13. Page 14 line 18. How was this aT dose chosen? Is it comparable to liver aT concentrations?
14. Page 15 line 6. This sentence is confusing. Is this in Table 1 or Fig 3?
15. Page 15 line 8. If comparing Fig 3a,b to Fig 3c, why is the Y-axis different?
16. Figure 3. Panel D, E and F should also have a heading of the cell type and the stimulant as in panel a,b,c
17. Figure 3. In panel f, why are the dots connected with lines? Are the cells incubated with both compounds and uptake measured? Clarify in the figure legend how this experiment was done and what the connecting lines mean.
18. Page 17 line 12-13 it is stated "Similar results were obtained for δ -TE-13'-COOH (Fig. 3d, e)." However, similar results were not found for d-TE-13-COOH because it reduced Ca but the a-T-13-COOH did not in Fig 3d.
19. Figure 5. aT-13-COOH needs statistical analysis as compared to w/o groups and not just compared to zieuton? Statistical comparison to the control nontreated group is very important for the conclusions being made. Also in Figure 5f, what inflammatory cell types are regulated by aT-13'-COOH?
20. Page 23 line 18. Selective accumulation of intracellular forms may also result in ability pass membranes or by lipid transport mechanisms. Are there differences in lipophilic properties of the metabolites?
21. Page 23 line 25. Something is wrong with the structure of this sentence. It doesn't make sense.
22. Figure 6. There is no analysis of inflammatory cells. Is there regulation of eosinophil accumulation in the lung or lung lavage?
23. Discussion page 25. This is a MAJOR CONCERN. The discussion implies that the effects of the

tocopherols/metabolites are only through these metabolites and inhibition of 5-LO, especially when discussing outcomes in clinical studies. However, the vitamin E forms/metabolites have other functions besides just an effect on 5LO, such as anti-oxidant activities and regulation of other enzymes including protein kinase C alpha. The discussion needs to encompass a broader more thorough view regarding vitamin E isoform/metabolite regulation of inflammation.

24. Page 31 line 21. Provide reference, source or site for MOE's pocket finder.

25. Page 32 line 2. Provide reference or source for GOLD 5.2 and for CHEMPLP.

26. Page 34 line 12. Provide reference for bronchial ring reactivity methods

27. Supplement figure 2. Why are only some of the tocopherols and metabolites shown in the docking? Clarify this in the figure legend.

Reviewer #2 (Remarks to the Author):

The present studies by Pein and colleagues present elucidate the biological actions of a new vitamin E-derived metabolite in the regulation of 5-LO activity and leukotriene biosynthesis. The experiments appear to be well conducted, the methods are well described and the statistical analysis conducted appear to be appropriate, furthermore the results support the authors' conclusions. This study is very interesting and potentially very important give both the clinical evidence and the wide spread use of vitamin E as anti-oxidant in food supplements. There are however several aspects that the authors need to address and they are as follows:

- 1) Vitamin E is widely used as antioxidant in fish oil supplements. The beneficial actions of these supplements are proposed to occur via the conversion of the omega-3 essential fatty acids to resolvins. Given that in the production of these molecules 5-LO plays an important role, the authors should discuss the impact of their findings on resolution processes.
- 2) It is interesting that the authors observed 12-LO activity in human neutrophils given that these cells do not express this enzyme to any appreciable levels. Can the authors comment on the origin of 12-LO products in these cells.
- 3) The authors state that circulating and exudate concentrations of alpha-T-13'-COOH are bioactive. To support this statement the authors should demonstrate that in activated peripheral blood from humans or in activated exudates from mice supplementation with Vitamin E at the same does used to measure the circulating metabolite concentrations also reduces 5-LO activity.
- 4) The authors demonstrate that in vitro leukocytes accumulate alpha-T-13'-COOH, does this happen also in vivo following vitamin E supplementation?
- 5) How do the concentrations of alpha-T-13'-COOH used in the in vivo models compare to the circulating and/or exudate concentrations?
- 6) Are cysteinyl leukotriene concentrations reduced following OVA challenge in mice given alpha-T-13'-COOH?
- 7) In several instances anti-inflammatories were shown to reduce short-term leukocyte recruitment but over the longer term inflammation was perturbed given that there was a shunting of the arachidonic acid to other pro-inflammatory pathways. The authors should investigate whether leukocyte recruitment as well as pro-inflammatory eicosanoid production remains reduced following alpha-T-13'-COOH administration at later intervals in the zymosan peritonitis model.

Reviewer #3 (Remarks to the Author):

This paper describes the preparation and screening of a library of vitamin E metabolites and semi-synthetic derivatives leading to the identification of long-chain ω -carboxylates as potent allosteric inhibitors of 5-lipoxygenase. Metabolism of α -tocopherol (using a liver-on-chip) gave a derivative, α -T-13-CO₂H, which was detected in human and mouse plasma at nM concentrations. UPLC-MS/MS was used to good effect and insights into the mode of binding gained. The metabolite suppresses inflammation and bronchial hyper-reactivity in mouse models of peritonitis and asthma. From this study it was concluded that the immune functions and protection against inflammation by α -tocopherol are in fact dependent upon the metabolite α -T-13-CO₂H which accumulates at sites of inflammation and selectively reduces 5-LO products in vitro and in vivo.

This is an interesting study which I believe will be of widespread interest. However I found the manuscript hard to read due, in part, to the use solely of abbreviations for the structures. I feel the paper would be improved by the use of structure numbers alongside the abbreviations for the various compounds (in the manuscript and supporting information). For the sake of readers not experts in the field, authors should ensure that abbreviations are explained as appropriate in the manuscript.

Page 5 and 6.....Conversion of α -tocopherol to α -T-13'-CO₂H is a key transformation and it would be helpful if this is shown with the full structures in one of the figures.

Page 7, line 5 ...do the authors mean complemented rather than completed?

I find many of the figures rather "busy"...the authors should consider if any of this information could be moved to the supplementary information.

It would be helpful in the supplementary information reporting the synthetic procedures if a scheme was given (starting material and product) rather than just the product as trivial names are used e.g. α -a-garcinoic acid.

The physical state of all products should be noted e.g. oil, crystalline solid etc.

In the data for α -T-13-CH₂OH, the authors should check the NMR data again carefully in particular δ 5.12 (dd, J 7.1, 15.8 Hz, 2-H) ...unlikely.

Reviewer #4 (Remarks to the Author):

The submitted manuscript by authors presents their results on the important role of a vitamin E long-chain metabolites (LCMs) against 5-LO-mediated formation of pro-inflammatory lipid mediators, by focusing on the 13-((2R)-6-hydroxy-2,5,7,8-tetramethyl-chroman-2-yl)-2,6,10-trimethyltridecanoic acid (α -T-13'-COOH). This particular metabolite is found to be the most potent inhibitor of cellular leukotriene biosynthesis among other LCMs. To this end, authors successfully prepared and screened the activity of a series of Vit-E metabolites, which are relevant to humans, against 5-LO-mediated leukotriene biosynthesis. They carried out a series of biological experiments and identified these long-chain carboxylates including α -T-13'-COOH as allosteric inhibitors of 5-LO, which is quite interesting, and can be of interest to the reader since it is not a common finding for 5-LO. Binding to 5-LO and direct interaction with 5-LO of the LCMs were also confirmed by a pull-down experiment using the amide coupling of the δ -TE-13'-COOH (another LCM metabolite with potent inhibitory activity against isolated 5-LO) to an amino-functionalized resin. Since the excess substrate (AA) neither effected the inhibition nor binding in pull-down experiments, the occurrence of a noncompetitive/allosteric

inhibition was suggested. Possibility of nuisance inhibition is also excluded. Authors predicted the binding location in a shallow cavity located between the catalytic domain and the regulatory C2-like domain based on molecular docking studies, which was supported by site-directed mutagenesis experiments.

They subsequently demonstrated in a human liver-on-a chip organoid that in fact α -T-13'-COOH is biosynthesized by human liver cells from α -tocopherol and is relevant to humans. Authors further proved that this metabolite is present at physiologically relevant concentrations in human and mouse plasma for effective inhibition of 5-LO-mediated formation of lipid mediators as well as it accumulates in immune cells and also in inflamed murine exudates. Lastly, authors showed that this special metabolite evidently elucidate in vivo activity by effectively suppressing inflammation in mouse peritonitis model and also suppressed the bronchial hyperactivity in mouse asthma model.

Overall the biological and chemical data are sound and presented in the manuscript in detail with statistical significance. The submitted manuscript is quite intriguing and meets the standards of Nature Communications. However, I have some minor comments:

- It seems to me that the authors made an emphasize on the two Vit-E metabolites throughout the manuscript for detailed biological experiments, namely δ -TE-13'-COOH and α -T-13'-COOH. α -T-13'-COOH is more effective in activated neutrophils (cellular assay) and δ -TE-13'-COOH is not. However, δ -TE-13'-COOH is found more effective on isolated 5-LO. Authors have discussed this difference in some parts of the manuscript. What I would like to know that if the minimum detectable limits for both δ -TE-13'-COOH and α -T-13'-COOH is the same (1 nM in Table 3 in Supp Info), were the authors still not able to detect at all the presence of the δ -TE-13'-COOH in plasma? So, is the δ -TE-13'-COOH not a physiologically relevant metabolite of Vit-E in humans? Can they elaborate on that?

- Addition of a Table with docking energies in the supporting section would be helpful for comparing the binding affinity of metabolites.

-It would be a lot easier to follow the manuscript if they give numbers to each metabolite in Tables and use this numbers next to the metabolite names throughout the manuscript.

-The J values are sometimes written italic and sometimes in regular fonts. They should be uniform.

Response to the reviewers

We gratefully thank the reviewers for their helpful comments which helped us to significantly improve the quality of our manuscript. In the following, we provide a detailed point-by-point list of the changes made (“answer”) in response to the referees’ suggestions (“comment”).

Reviewer 1

Comment: This report demonstrates that metabolites of alpha-tocopherol inhibit 5-LO. The biochemistry analysis for these interactions is outstanding and statistics are appropriate. Moreover, this is a very important contribution to the field.

Answer: We thank the reviewer for this kind evaluation of our work.

Comment: 1. Page 4 line 5-6. What forms of vitamin E? Indicate the isoforms when discussing studies in the introduction rather than using the term vitamin E. Vitamin E is used to incorrectly indicate alpha-tocopherol when vitamin E really is a general term encompassing all the isoforms.

Answer: We addresses this comment of the reviewer accordingly; α -tocopherol and other vitamin E forms are now indicated throughout the introduction, and the term “vitamin E” is only used when all forms are addressed.

Comment: 2. Page 4 line 15. Delete the word "Thus". This sentence is not a conclusion of the previous sentence. It makes it confusing.

Answer: Done.

Comment: 3. Page 7 line 6. Indicate where in the supplementary information because it is at the end and hard to find. Indicate something like “supplementary note in supplement information”

Answer: Done, we now write “Supplementary Note 1” in the revised manuscript in line with the formatting instructions of *Nature Communications*.

Comment: 4. Fig 1e should have a more thorough description of the length of times for first concentration and diluted concentration and when substrate was present.

Answer: Done, pre-incubation times and the time point when the reaction was started by addition of substrate is now given in the figure legend. A more detailed description of the experimental procedure can be found in the Method section (page 24, line 25 – page 25, line 4).

Comment: 5. For Fig 1f, as a control to demonstrate specificity, include competition with increasing concentrations of non-labeled compounds.

Answer: We thank the reviewer for this point which is well taken. Competition studies with the free compound may indeed help to verify the specificity of target-ligand interactions. We here decided to exclude such an experiment, which would be very laborious and time-consuming because of the high amount of natural product required for synthesizing the δ -TE-13'-COOH-coupled construct. Another complication might arise from the poor water solubility of δ -TE-13'-COOH, which should be given in excess reaching high micromolar concentrations. To circumvent these difficulties, we addressed the functional interaction of δ -TE-13'-COOH with 5-lipoxygenase by analyzing the conversion of the substrate arachidonic acid through the purified enzyme in a cell-free assay. This activity assay recognizes functional enzyme-ligand interactions while the binding assay cannot discriminate between functional and non-functional binding and thus only provides limited insights. We hope that the reviewer agrees with our decision.

Comment: 6. In the legend of Fig 1, why is the 4-phenoxy butyric acid the control? This is confusing. Provide rationale.

Answer: We agree with the reviewer that we have to better explain this aspect in the revised manuscript, and we accordingly revised the manuscript to address the reviewer's concern. The commercially available 4-phenoxy butyric acid represents a highly truncated vitamin E metabolite, which maintains the carboxylic acid function (which is essential for amide coupling) and the phenoxy moiety as part of the chromanol ring system. The aliphatic side chain is shortened, the hydroxyl group removed, and the chromanol ring disrupted. The rationale for selecting 4-phenoxy butyric acid as structurally related, inactive control is now mentioned in the figure legend. Additional controls with closely related inactive vitamin E metabolites were not possible because of insufficient amounts of isolated and semi-synthetic compounds. Natural forms of vitamin E could also not be used because they lack a functional group at the side chain, which is required for appropriate coupling to amino-functionalized sepharose.

Comment: 7. Page 9 line 10. d-TE-13'-COOH compound is not in Fig. 1b but DE-13'-COOH is in the figure. Add d-TE-13'-COOH and its IC50 to Fig 1b.

Answer: Done. According to the formatting instructions, Fig. 1b is now displayed as Table 2.

Comment: 8. Page 9 line 11. Nuisance is vague. Does this mean nonspecific? background?....

Answer: Yes, we meant compounds, which form colloid-like aggregates and inhibit the target without specificity and with flat SARs. Non-ionic detergents such as Triton X-100 efficiently reduce aggregate formation by lipophilic/amphiphilic compounds. The term nuisance inhibition is now defined on page 7, lines 17-19.

9. Page 9 line 27. What is meant by "retain" in this sentence?

Answer: The sentence has been rewritten and now clearly states that "... bulky amides such as δ -TE-13'-CONHBz (7) inhibit 5-LO comparably to acidic LCMs".

Comment: 10. Page 10 text. For clarity, use the same AA nomenclature in the text as that in the figure 1i.

Answer: Done.

Comment: 11. Figure 2. panel e and f are confusing as to how they are different than panel e and d. It needs to be more clearly labeled in the figure that e and f have a-T in both groups per panel.

Answer: Done, the groups in panel e and f are now labeled as " α -T" and " α -T + Ses.", where Ses. is the defined abbreviation for sesamin.

Comment: 12. Page 13 line 25. In the methods aT is given either i.p. or orally. In the figure legends it is not clearly indicated whether the tocopherol in each panel is either i.p. or orally administered.

Answer: Thank you for this comment. For better clarity, we have rearranged the legend of Fig. 2 and now indicate next to the dose that α -T is given p.o. [" α -T (50 mg/kg, p.o.)"].

Comment: 13. Page 14 line 18. How was this aT dose chosen? Is it comparable to liver aT concentrations?

Answer: Yes, the concentration of α -T (100 μ M) was chosen as an intermediate between human plasma level (14 - 27 μ M) and hepatic tissue concentrations (70 - 530 nmol/g) (Nagita et al., 1997, Hepatology, 26(2): 392); this is now mentioned in the text (page 12, line 3).

Comment: 14. Page 15 line 6. This sentence is confusing. Is this in Table 1 or Fig 3?

Answer: Done, the link to Table 1 has now been placed at the end of the sentence because also the second part refers to the table.

Comment: 15. Page 15 line 8. If comparing Fig 3a,b to Fig 3c, why is the Y-axis different?

Answer: In panel a and b, the 5-LO products LTB₄, its isomers and 5-H(P)ETE were determined by HPLC. Since activated monocytes synthesize considerably less 5-LO products than PMNL, we applied our more sensitive UPLC-MS/MS method to analyze LTB₄ (panel c). The amounts of 5-LO products were not summarized for UPLC-MS/MS analysis because our UPLC-MS/MS method has been optimized for the detection of changes and not for absolute quantitation. The different analytic methods used are now specified in the figure legend.

Comment: 16. Figure 3. Panel D, E and F should also have a heading of the cell type and the stimulant as in panel a,b,c

Answer: Done.

Comment: 17. Figure 3. In panel f, why are the dots connected with lines? Are the cells incubated with both compounds and uptake measured? Clarify in the figure legend how this experiment was done and what the connecting lines mean.

Answer: Done, we now describe this better (page 50, line 25 – page 51, line 1). PMNL were incubated with either δ -TE-13'-COOH or α -T-13'-COOH. The connected lines indicate data from the same independent experiment.

Comment: 18. Page 17 line 12-13 it is stated "Similar results were obtained for δ -TE-13'-COOH (Fig. 3d, e)." However, similar results were not found for d-TE-13-COOH because it reduced Ca but the a-T-13-COOH did not in Fig 3d.

Answer: Thank you for highlighting this issue; we now state: “While similar results were obtained for δ -TE-13'-COOH (**4h**) at submicromolar concentrations, intracellular Ca^{2+} -influx was significantly reduced at $\geq 1 \mu\text{M}$ ”.

Comment: 19. Figure 5. α T-13-COOH needs statistical analysis as compared to w/o groups and not just compared to zileuton? Statistical comparison to the control nontreated group is very important for the conclusions being made.

Answer: Statistical analysis has been performed by one-way ANOVA + Tukey HSD post-hoc tests. All groups were compared to the control (w/o) group. To enhance the readability of the figure, the statistical significance (e.g., of α -T-13'-COOH vs. w/o) is expressed using asterisks (* $p < 0.05$, ** $p < 0.01$, *** $p < 0.001$) as indicated in the legend. Additional comparisons between groups (e.g., α -T-13'-COOH vs. zileuton) depend on two-tailed unpaired student t-tests and are indicated by vertical bars.

Comment: Also in Figure 5f, what inflammatory cell types are regulated by α T-13'-COOH?

Answer: Zymosan injection stimulates peritoneal macrophages that are resident in the peritoneal cavity to produce chemotactic LTB_4 . Influx of leukocytes, predominantly neutrophils, peaks at 4 hours after peritonitis induction, as shown by us and others (Doherty et al., 1985, *Prostaglandins*, 30: 769; Rao et al., 1994, *J. Pharmacol. Exp. Ther.*, 269: 917; Rossi et al., 2014, *Pharmacol. Res.*, 87: 1; Pace et al., 2017, *Sci. Rep.*, 7: 3759; Hanke et al., 2013, *J. Med. Chem.*, 56: 9031). We now mention that neutrophils are the major infiltrating cell type on page 15, line 4.

Comment: 20. Page 23 line 18. Selective accumulation of intracellular forms may also result in ability pass membranes or by lipid transport mechanisms. Are there differences in lipophilic properties of the metabolites?

Answer: Thank you, this is a very interesting question, which we are currently exploring in detail. As now discussed on page 17, lines 18-21, α -T-13'-COOH is more lipophilic than δ -TE-13'-COOH (calculated log P values: 6.65 vs. 8.81). Accordingly, the retention time of α -T-13'-COOH on a C18-RP column is delayed compared to δ -TE-13'-COOH (2.7 min vs. 1.5 min).

Comment: 21. Page 23 line 25. Something is wrong with the structure of this sentence. It doesn't make sense.

Answer: Agreed; this sentence has been rewritten for better clarity and reads now as follows: “Multiple LCMs (including δ -TE-13'-COOH, **4h**) inhibit 5-LO but only α -T-13'-COOH (**4a**; ²⁷ and this study), α -T-13'-CH₂OH (**2a**)²⁹ and chain-shortened ω -carboxylates of α -T (**1a**; this study) have been detected in humans and rodents. The physiological relevance of non-detected metabolites remains speculative in light of comparable detection limits for long-chain ω -carboxylates (Supplementary Table 2).”

Comment: 22. Figure 6. There is no analysis of inflammatory cells. Is there regulation of eosinophil accumulation in the lung or lung lavage?

Answer: In this particular experiments we did not analyze the infiltration of inflammatory cells but only addressed bronchial hyper-reactivity as an important hallmark of asthma, which is strongly associated to leukotriene production (Roviezzo et al., 2017, *Front. Pharmacol.*, 8, 857; Rossi et al., 2017, *Front. Pharmacol.*, 7: 525; Schaible et al., 2016, *Biochem. Pharmacol.*, 112, 60). In fact, we now show that α -T-13'-COOH lowers LTC₄ to basal levels (new Fig. 6b). Follow-up studies are planned to investigate additional asthma-like features including pulmonary inflammation and eosinophil infiltration. However, these experiments require a new ethical approval (not possible within the given timeframe) and the results will be published separately.

Comment: 23. Discussion page 25. This is a MAJOR CONCERN. The discussion implies that the effects of the tocopherols/metabolites are only through these metabolites and inhibition of 5-LO, especially when discussing outcomes in clinical studies. However, the vitamin E forms/metabolites have other functions besides just an effect on 5LO, such as anti-oxidant activities and regulation of other enzymes including protein kinase C alpha. The discussion needs to encompass a broader more thorough view regarding vitamin E isoform/metabolite regulation of inflammation.

Answer: Point well taken by the referee and done. The multiple activities of vitamin E forms and metabolites are now discussed in light of their potential contribution to the anti-inflammatory activity of vitamin E on page 19, paragraph 2.

Comment: 24. Page 31 line 21. Provide reference, source or site for MOE's pocket finder.

Answer: Done, the site for MOE's pocket finder is now given as URL in the text (page 27, lines 7-8).

Comment: 25. Page 32 line 2. Provide reference or source for GOLD 5.2 and for CHEMPLP.

Answer: Done, the source for GOLD 5.2 and a site for CHEMPLP are now given in the text (page 27, lines 12-14). The reference for ChemPLP is given in Supplementary Table 1.

Comment: 26. Page 34 line 12. Provide reference for bronchial ring reactivity methods

Answer: Done, see page 29, lines 21-22.

Comment: 27. Supplement figure 2. Why are only some of the tocopherols and metabolites shown in the docking? Clarify this in the figure legend.

Answer: We have extended Supplementary Fig. 2 and now show docking poses for all tocopherols, tocotrienols, LCMs and derivatives for which we have biological data.

Reviewer 2

Comment: The present studies by Pein and colleagues elucidate the biological actions of a new vitamin E-derived metabolite in the regulation of 5-LO activity and leukotriene biosynthesis. The experiments appear to be well conducted, the methods are well described and the statistical analysis conducted appear to be appropriate, furthermore the results support the authors' conclusions. This study is very interesting and potentially very important give both the clinical evidence and the wide spread use of vitamin E as anti-oxidant in food supplements.

Answer: Thank you for this favorable feedback.

Comment: 1) Vitamin E is widely used as antioxidant in fish oil supplements. The beneficial actions of these supplements are proposed to occur via the conversion of the omega-3 essential fatty acids to resolvins. Given that in the production of these molecules 5-LO plays an important role, the authors should discuss the impact of their findings on resolution processes.

Answer: Thank you for this valuable comment! We now discuss this important point on page 20, line 22 – page 21, line 8.

Comment: 2) It is interesting that the authors observed 12-LO activity in human neutrophils given that these cells do not express this enzyme to any appreciable levels. Can the authors comment on the origin of 12-LO products in these cells.

Answer: The 12-LO activity in our PMNL preparations likely derives from platelet contaminations (that can be hardly circumvented); this is now mentioned on page 25, lines 21-22.

Comment: 3) The authors state that circulating and exudate concentrations of alpha-T-13'-COOH are bioactive. To support this statement the authors should demonstrate that in activated peripheral blood from humans or in activated exudates from mice supplementation with Vitamin E at the same dose used to measure the circulating metabolite concentrations also reduces 5-LO activity.

Answer: Done, the new Fig. 3d now shows that α -T-13'-COOH (100 nM) inhibits LTB₄ formation in LPS/fMLP-stimulated human whole blood.

Comment: 4) The authors demonstrate that in vitro leukocytes accumulate alpha-T-13'-COOH, does this happen also in vivo following vitamin E supplementation?

Answer: Thank you for this important point that we have addressed by performing additional experiments. Our new data show that α -T-13'-COOH, α -T-11'-COOH and α -T-9'-COOH reach substantially higher concentrations in leukocytes that were isolated from peritoneal exudates of zymosan-challenged mice as compared to the exudate (see new Figs. 2e and 3h and Supplementary Fig. 3). Please note that we had to prolong the treatment of mice with zymosan from 30 min to at least 4 hours to obtain sufficient leukocytes from the peritoneal exudate for further analysis. The concentration of α -T-13'-COOH markedly decreased in the exudate during this period (compare Fig. 2d and e), and oral administration of α -tocopherol (single or repeated administration) only enhanced the accumulation of α -T-13'-COOH in peritoneal leukocytes when given prior to cell influx (Figs. 2e and 3h and Supplementary Fig. 3).

Comment: 5) How do the concentrations of alpha-T-13'-COOH used in the in vivo models compare to the circulating and/or exudate concentrations?

Answer: α -T-13'-COOH reaches concentrations of 1.4 μ M in plasma and 3.9 μ M in the peritoneal exudate when i.p. administered to mice at a dose of 10 mg/kg (Fig. 5b). Plasma and exudate concentrations upon oral administration of α -T were more than 15-times lower (Fig. 2c and d). This difference is now mentioned on page 14, line 25 – page 15, line 2. Further *in vivo* studies determining the circulating and lung tissue concentrations of α -T-13'-COOH in ovalbumin-induced bronchial hyperreactivity are in progress but will be published separately.

Comment: 6) Are cysteinyl leukotriene concentrations reduced following OVA challenge in mice given α -T-13'-COOH?

Answer: Yes, our new results show that α -T-13'-COOH reduces LTC₄ to basal levels at day 21 (new Fig. 6b) when bronchial reactivity was assessed (Fig. 6c).

Comment: 7) In several instances anti-inflammatories were shown to reduce short-term leukocyte recruitment but over the longer term inflammation was perturbed given that there was a shunting of the arachidonic acid to other pro-inflammatory pathways. The authors should investigate whether leukocyte recruitment as well as pro-inflammatory eicosanoid production remains reduced following α -T-13'-COOH administration at later intervals in the zymosan peritonitis model.

Answer: Thank you for this interesting question, which we addressed by additional experiments. α -T-13'-COOH (10 mg/kg) and the reference 5-LO inhibitor zileuton (10 mg/kg) were given i.p. to mice 30 min before peritonitis induction. Animals were sacrificed 18 hours (1080 min) after zymosan injection, which is the latest time point before peritoneal cell numbers readjust to baseline in this self-resolving model (see Fig. below; Troisi et al., 2016, unpublished data).

Fig.: Zymosan-induced peritonitis in mice. Time-dependent changes of cell numbers in the peritoneal cavity upon injection of zymosan.

While immune cell infiltration into the peritoneal cavity was substantially reduced by both α -T-13'-COOH and zileuton at 4 hours post zymosan injection (Fig. 5f, left panel), the two 5-LO inhibitors were without effect at the late time point (18 h) as is now shown in the new Fig. 5f (right panel). Along these lines, α -T-13'-COOH and zileuton slightly elevated the levels of diverse lipid mediators such as pro-inflammatory COX- and 5-LO products in the exudate, but pro-resolving protectins and maresins were also increased (Fig. 5i and Supplementary Fig. 6). Note that the concentrations of pro-inflammatory lipid mediators were strongly decreased at 18 hours compared to 4 hours after zymosan injection (see Fig. 5e, h and i and Supplementary Fig. 6), and the physiological relevance of these low levels is questionable. On the other hand,

α -T-13'-COOH (but not zileuton) strongly increased the circulating concentration of resolvin E3 in plasma (Fig. 5j) while the levels of other lipid mediators were either decreased or remained unchanged (Supplementary Fig. 6), which might hint towards a role of α -T-13'-COOH in systemic resolution. This is now discussed on page 15, last paragraph).

Reviewer 3

Comment: This paper describes the preparation and screening of a library of vitamin E metabolites and semi-synthetic derivatives leading to the identification of long-chain ω -carboxylates as potent allosteric inhibitors of 5-lipoxygenase. Metabolism of α -tocopherol (using a liver-on-chip) gave a derivative, α -T-13-CO₂H, which was detected in human and mouse plasma at nM concentrations. UPLC-MS/MS was used to good effect and insights into the mode of binding gained. The metabolite suppresses inflammation and bronchial hyper-reactivity in mouse models of peritonitis and asthma. From this study it was concluded that the immune functions and protection against inflammation by α -tocopherol are in fact dependent upon the metabolite α -T-13-CO₂H which accumulates at sites of inflammation and selectively reduces 5-LO products in vitro and in vivo. This is an interesting study which I believe will be of widespread interest.

Answer: We kindly thank the reviewer for this comprehensive and positive evaluation.

Comment: However I found the manuscript hard to read due, in part, to the use solely of abbreviations for the structures. I feel the paper would be improved by the use of structure numbers alongside the abbreviations for the various compounds (in the manuscript and supporting information). For the sake of readers not experts in the field, authors should ensure that abbreviations are explained as appropriate in the manuscript.

Answer: We agree with the reviewer and accordingly revised the manuscript to address this issue. We have introduced numbers to label vitamin E forms, metabolites and derivatives next to the abbreviations of the vitamin E nomenclature system throughout the text, tables and figures.

Comment: Page 5 and 6.....Conversion of α -tocopherol to α -T-13'-CO₂H is a key transformation and it would be helpful if this is shown with the full structures in one of the figures.

Answer: Done, the conversion of α -tocopherol to α -T-13'-COOH in liver is now shown in Supplementary Fig. 1b and full structures are provided. We have added the labels ' α -T (1a)'

and ' α -T-13'-COOH (**4a**)' to the structures and now also provide a link to this figure on page 6, line 2.

Comment: Page 7, line 5 ...do the authors mean complemented rather than completed?

Answer: Yes, we exchanged "completed" by "complemented".

Comment: I find many of the figures rather "busy"...the authors should consider if any of this information could be moved to the supplementary information.

Answer: Agreed, the old Fig. 2g was moved to the Supplementary information and now reads as Supplementary Figure 4c. Moreover, the new Supplementary Figs. 3 and 6 as well as Supplementary Tables 1 and 5 were included into the Supplementary information.

Comment: It would be helpful in the supplementary information reporting the synthetic procedures if a scheme was given (starting material and product) rather than just the product as trivial names are used e.g. α -garcinoic acid.

Answer: Done, schemes have been drawn as heading for each synthetic procedure.

Comment: The physical state of all products should be noted e.g. oil, crystalline solid etc.

Answer: Done, the information about the physical state has been added when missing.

Comment: In the data for α -T-13-CH₂OH, the authors should check the NMR data again carefully in particular d5.12 (dd, J 7.1, 15.8 Hz, 2-H) ...unlikely.

Answer: Done, multiplicity of this particular signal has been corrected and changed for a multiplet for two vinylic protons.

Reviewer 4

Comment: The submitted manuscript by authors presents their results on the important role of a vitamin E long-chain metabolites (LCMs) against 5-LO-mediated formation of pro-inflammatory lipid mediators, by focusing on the 13-((2R)-6-hydroxy-2,5,7,8-tetramethylchroman-2-yl)-2,6,10-trimethyltridecanoic acid (α -T-13'-COOH). This particular metabolite is found to be the most potent inhibitor of cellular leukotriene biosynthesis among other LCMs.

To this end, authors successfully prepared and screened the activity of a series of Vit-E metabolites, which are relevant to humans, against 5-LO-mediated leukotriene biosynthesis. They carried out a series of biological experiments and identified these long-chain carboxylates including α -T-13'-COOH as allosteric inhibitors of 5-LO, which is quite interesting, and can be of interest to the reader since it is not a common finding for 5-LO. Binding to 5-LO and direct interaction with 5-LO of the LCMs were also confirmed by a pull-down experiment using the amide coupling of the δ -TE-13'-COOH (another LCM metabolite with potent inhibitory activity against isolated 5-LO) to an amino-functionalized resin. Since the excess substrate (AA) neither effected the inhibition nor binding in pull-down experiments, the occurrence of a noncompetitive/allosteric inhibition was suggested. Possibility of nuisance inhibition is also excluded. Authors predicted the binding location in a shallow cavity located between the catalytic domain and the regulatory C2-like domain based on molecular docking studies, which was supported by site-directed mutagenesis experiments. They subsequently demonstrated in a human liver-on-a chip organoid that in fact α -T-13'-COOH is biosynthesized by human liver cells from α -tocopherol and is relevant to humans. Authors further proved that this metabolite is present at physiologically relevant concentrations in human and mouse plasma for effective inhibition of 5-LO-mediated formation of lipid mediators as well as it accumulates in immune cells and also in inflamed murine exudates. Lastly, authors showed that this special metabolite evidently elucidate in vivo activity by effectively suppressing inflammation in mouse peritonitis model and also suppressed the bronchial hyperactivity in mouse asthma model. Overall the biological and chemical data are sound and presented in the manuscript in detail with statistical significance. The submitted manuscript is quite intriguing and meets the standards of Nature Communications.

Answer: Many thanks for your careful evaluation and encouraging assessment.

Comment: It seems to me that the authors made an emphasize on the two Vit-E metabolites throughout the manuscript for detailed biological experiments, namely δ -TE-13'-COOH and α -T-13'-COOH. α -T-13'-COOH is more effective in activated neutrophils (cellular assay) and δ -TE-13'-COOH is not. However, δ -TE-13'-COOH is found more effective on isolated 5-LO. Authors have discussed this difference in some parts of the manuscript. What I would like to know that if the minimum detectable limits for both δ -TE-13'-COOH and α -T-13'-COOH is the same (1 nM in Table 3 in Supp Info), were the authors still not able to detect at all the presence of the δ -TE-13'-COOH in plasma? So, is the δ -TE-13'-COOH not a physiologically relevant metabolite of Vit-E in humans? Can they elaborate on that?

Answer: In fact, neither δ -TE-13'-COOH nor δ -DE-13'-COOH (the proposed physiological metabolite of δ -TE according to *in vitro* studies; Freiser et al., 2009, *J. Nutr.*, 139:884) were

detected in human and mouse plasma. The extracted chromatogram for δ -TE-13'-COOH in female plasma is exemplarily shown below in comparison to α -T-13'-COOH.

Human female plasma: extracted chromatograms for α -T-13'-COOH and δ -TE-13'-COOH

δ -TE-13'-COOH is therefore unlikely to be a relevant bioactive vitamin E metabolite at Western standard diet which is rich in α -T but only contains low quantities of δ -TE. This discrepancy is further enhanced because the liver actively sorts out α -T among resorbed vitamin E forms. α -T is then preferentially secreted within plasma lipoproteins into the bloodstream and reaches considerably higher plasma levels than δ -T in humans (25-35 μ M vs. 0.01 μ M) (Mangialasche et al., 2012, *Neurobiol. Aging*, 33: 2282). On the other hand, our data does not exclude a physiological relevance of δ -TE-13'-COOH / δ -DE-13'-COOH in other tissues, e.g., in liver where vitamin E metabolites are strongly enriched (Jiang et al., 2007, *Lipid Res.*, 48:1221) or under different dietary conditions. Along these lines, δ -TE is efficiently metabolized by hepatocarcinoma HepG2 and lung carcinoma A549 cells (You et al., 2005, *J. Nutr.*, 135:227; Galli et al., 2017, *Free Rad. Biol. Med.*, 102:16), and γ -TE-13'-COOH levels strongly increase up to 140 nM in rat plasma and 730 nM in rat liver when γ -TE is orally given at pharmacological doses (Jiang et al., 2007, *Lipid Res.*, 48:1221). These points are now discussed on page 18, first paragraph.

Comment: Addition of a Table with docking energies in the supporting section would be helpful for comparing the binding affinity of metabolites.

Answer: Done, docking scores are now listed in the new Supplementary Table 1. We need to point out that the calculated docking energies are estimates of binding affinities and do not necessarily correlate with the measured biological activities of the compounds. Our inhibitor studies on 5-LO reflect an impact on the functionality of the enzyme and not direct binding (as a radioligand binding assay would do).

Comment: It would be a lot easier to follow the manuscript if they give numbers to each metabolite in Tables and use this numbers next to the metabolite names throughout the manuscript.

Answer: Done, see also first comment of reviewer 3.

Comment: The J values are sometimes written italic and sometimes in regular fonts. They should be uniform.

Answer: Done, the format of the NMR data has been carefully checked and unified.

REVIEWERS' COMMENTS:

Reviewer #1 (Remarks to the Author):

Overall this is an outstanding comprehensive study and it is presented well. The authors have addressed all the concerns of the reviewers. Within the revisions, on page 21 line 3, it is mentioned that fish oil contains aT. However, often fish oil contains gT rather than aT as an anti-oxidant. This should be addressed.

Reviewer #2 (Remarks to the Author):

The authors had address my queries in full and the manuscript is substantially improved.

Reviewer #4 (Remarks to the Author):

I have carefully investigated the responses of authors to reviewers' comments point by point. First of all, this manuscript provides an outstanding work demonstrating that certain metabolites of alpha-tocopherol inhibit 5-LO, which may contribute its pharmacological effect. It is certainly a very important contribution to the field.

When I checked the revised manuscript, I see that the authors successfully responded and completed the all important revisions raised by reviewers, by either inserting explanatory comments in the main text or by providing additional experimental evidences. All revisions made by the authors were highlighted in the body of text making easy to follow point by point.

All of the questions raised by reviewers' were satisfactorily responded by authors and the new version of manuscript is more easy to follow and understandable with a high scientific quality.

In my opinion, the revised manuscript is acceptable as it is for publication.

Response to the reviewers

We are very happy about the positive feedback of the reviewers and want to thank them again for their valuable comments throughout the revision process. In the following, we provide a detailed point-by-point list of the changes made (“answer”) in response to the referees’ points (“comment”).

Reviewer 1

Comment: Overall this is an outstanding comprehensive study and it is presented well. The authors have addressed all the concerns of the reviewers.

Answer: We are grateful for this favorable evaluation.

Comment: Within the revisions, on page 21 line 3, it is mentioned that fish oil contains aT. However, often fish oil contains gT rather than aT as an anti-oxidant. This should be addressed.

Answer: Thank you for addressing this interesting point. We now write on page 21, lines 3-4: “This aspect has particular relevance for fish oil supplements where tocopherols, including α -T (1a) and γ -T (1c), are added as an anti-oxidant ...”

Reviewer 2

Comment: The authors had address my queries in full and the manuscript is substantially improved.

Answer: Many thanks for the positive assessment.

Reviewer 4

Comment: I have carefully investigated the responses of authors to reviewers' comments point by point. First of all, this manuscript provides an outstanding work demonstrating that certain metabolites of alpha-tocopherol inhibit 5-LO, which may contribute its pharmacological effect. It is certainly a very important contribution to the field. When I checked the revised manuscript, I see that the authors successfully responded and completed the all important revisions raised by reviewers, by either inserting explanatory comments in the main text or by providing additional experimental evidences. All revisions made by the authors were highlighted in the body of text making easy to follow point by point. All of the questions raised by reviewers' were satisfactorily responded by authors and the new version of manuscript is more easy to follow

and understandable with a high scientific quality. In my opinion, the revised manuscript is acceptable as it is for publication.

Answer: We are delighted by the comprehensive and kind evaluation of our revised work.